# Real-Time Motion Prediction via Heterogeneous Polyline Transformer with Relative Pose Encoding

**Zhejun Zhang**
Computer Vision Lab
ETH Zurich
Zurich, Switzerland
zhejun.zhang@vision.ee.ethz.ch

**Alexander Liniger**
Computer Vision Lab
ETH Zurich
Zurich, Switzerland
alex.liniger@vision.ee.ethz.ch

**Christos Sakaridis**
Computer Vision Lab
ETH Zurich
Zurich, Switzerland
csakarid@vision.ee.ethz.ch

**Fisher Yu**
Computer Vision Lab
ETH Zurich
Zurich, Switzerland
i@yf.io

**Luc Van Gool**
CVL, ETH Zurich, CH
PSI, KU Leuven, BE
INSAIT, Un. Sofia, BU
vangool@vision.ee.ethz.ch

## Abstract

The real-world deployment of an autonomous driving system requires its components to run on-board and in real-time, including the motion prediction module that predicts the future trajectories of surrounding traffic participants. Existing agent-centric methods have demonstrated outstanding performance on public benchmarks. However, they suffer from high computational overhead and poor scalability as the number of agents to be predicted increases. To address this problem, we introduce the K-nearest neighbor attention with relative pose encoding (KNARPE), a novel attention mechanism allowing the pairwise-relative representation to be used by Transformers. Then, based on KNARPE we present the Heterogeneous Polyline Transformer with Relative pose encoding (HPTR), a hierarchical framework enabling asynchronous token update during the online inference. By sharing contexts among agents and reusing the unchanged contexts, our approach is as efficient as scene-centric methods, while performing on par with state-of-the-art agent-centric methods. Experiments on Waymo and Argoverse-2 datasets show that HPTR achieves superior performance among end-to-end methods that do not apply expensive post-processing or model ensembling. The code is available at https://github.com/zhejz/HPTR.

## 1 Introduction

Motion prediction is an important component of modular autonomous driving stack [63]. As the downstream module of perception [22, 42, 48] and the upstream module of planning [3, 9, 39], the task of motion prediction [16, 60] is to predict the multi-modal future trajectories of other agents next to the self-driving vehicle (SDV) based on the heterogeneous observations, including for example high-definition (HD) maps, traffic lights and other vehicles, pedestrians and cyclists. This is an essential task because without accurate and real-time prediction results [30], the planning module cannot safely and comfortably navigate the SDV through highly interactive driving environments.

To achieve top performance on public motion prediction leaderboards [1, 58], state-of-the-art (SOTA) methods [36, 47, 52, 55] leverage agent-centric vectorized representations and Transformer-based network architectures. However, the good performance of these approaches comes at a cost of high computational overhead as illustrated in Figure 1. The most well-known problem of agent-centric

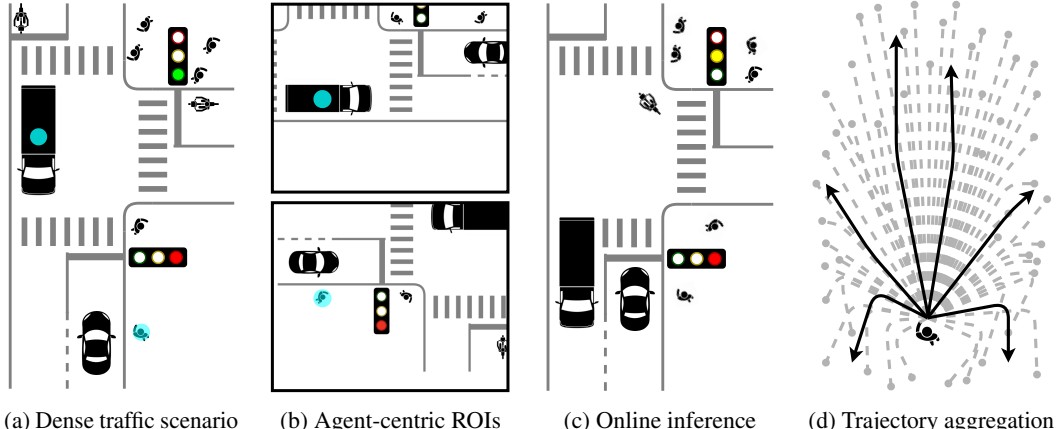

(a) Dense traffic scenario     (b) Agent-centric ROIs     (c) Online inference     (d) Trajectory aggregation

Figure 1: To efficiently predict the multi-modal future of numerous agents in dense traffic (1a), HPTR minimizes the computational overhead by: (1b) Sharing contexts among target agents. (1c) Reusing static contexts during online inference. (1d) Avoiding expensive post-processing and ensembling.

approaches is the poor scalability as the number of target agents grows in urban driving environments with dense traffic; for example the busy intersection in Figure 1a. Although the agent-centric regions of interest (ROI) in Figure 1b are largely overlapping, the same context is transformed to the coordinate system of each target agent and independently saved and processed. This causes a huge waste of computational resources, which is often neglected by prior works because their experiments focus on offline inference that queries the prediction only once given a scenario from the dataset. In contrast to offline inference, the motion prediction module of a real-world SDV is continuously queried with streaming inputs during online inference. For example, in Figure 1c, the prediction module is queried again shortly after Figure 1a. Although most contexts remain unchanged after this short period of time, existing methods would start inference from scratch without reusing the encoded features of static contexts. Moreover, prior works often predict a massive number of redundant trajectories and ensemble the outputs of numerous models, such that the final output can be adjusted in favor of prediction diversity during the post-processing, as illustrated in Figure 1d. Although these techniques significantly boost the performance on public benchmarks, they should be sparingly used on a real-world SDV where the computational resources are scarce and the raw predictions, rather than the heuristically aggregated ones, are preferred by the downstream planning module.

To address these problems, our first step is to represent everything as heterogeneous polylines [17]. Then we adopt the pairwise-relative representation [11, 28] and separate the local attribute from the global pose of each polyline. The local attribute specifies what the polyline actually is and it will be shared or reused if possible. The global pose specifies where the polyline is and it will be used to derive the pairwise-relative pose before being processed by the neural networks. Considering polylines as tokens, our second step is to introduce the **K**-nearest **N**eighbor **A**ttention with **R**elative **P**ose **E**ncoding (KNARPE) module, which allows Transformers [53] to aggregate the local contexts for each token via the local attributes and relative poses. While KNARPE enables context sharing among agents, we further propose the **H**eterogeneous **P**olyline **T**ransformer with **R**elative pose encoding (HPTR), which emphasizes the heterogeneous nature of polylines in motion prediction tasks. Based on KNARPE, HPTR uses hierarchical Transformer encoders and decoders to separate the intra-class and inter-class attention, such that tokens can be updated asynchronously during online inference. More specifically, for static polylines, such as HD maps, their features will be reused, whereas for dynamic polylines, such as traffic lights and agents, their features will be updated on demand.

Our contributions are summarized as follows: (1) We introduce KNARPE, a novel attention mechanism that enables the pairwise-relative representation to be used by Transformer-based architectures. (2) Based on KNARPE we propose HPTR, a hierarchical framework that minimizes the computational overhead via context sharing among agents and asynchronous token update during online inference. (3) Compared to SOTA agent-centric methods, we achieve similar performance while reducing the memory consumption and the inference latency by 80%. By caching the static map features during online inference, HPTR can generate predictions for 64 agents in real time at 40 frames per second. Experiments on Waymo and Argoverse-2 datasets show that our approach compares favorably against other end-to-end methods which do not apply expensive post-processing and ensembling.

## 2 Related work

**Motion prediction** is a popular research topic because it is an essential component of modular autonomous driving stacks [63]. The task of motion prediction has been formulated in different ways. In this paper, we focus on the most popular one: the marginal motion prediction [6, 8, 27, 32, 33] where the multi-modal future trajectories are predicted individually for each target agent [1, 38, 58]. In contrast to the marginal formulation, joint motion prediction [19, 20, 35, 49] requires the multi-modal futures to be simultaneously generated for all target agents from the same scenario [2, 57, 66]. Beyond the open-loop motion prediction tasks, behavior simulation [44, 50, 68] is formulated in a closed-loop fashion where the future agent trajectories are simulated by rolling out a learned policy [3, 59]. Numerous works have also investigated the possibility to combine motion prediction with other modules, such as perception and planning [7, 15, 23, 29], or to learn an end-to-end driving policy [24, 25, 67] mapping sensor measurements directly to the actions or motion plans of the SDV.

**Vectorized representation** proposed by VectorNet [17] is widely used in recent works because of its promising performance [47, 52, 55]. Depending on how the coordinate system is selected, the vectorized representation falls into three categories: agent-centric, scene-centric and pairwise-relative. The most popular one is the agent-centric representation [36, 47, 52, 55] that transforms all inputs to the local coordinates of each target agent. Despite its good performance, this approach suffers from poor scalability as the number of target agents grows. To reduce the computational overhead, scene-centric representation [37] shares the contexts among all target agents by transforming all inputs to a global coordinate system, which is not tied to any specific agent but to the whole scene. However, its performance is poor due to the lack of rotation and translation invariance. In this paper we use the pairwise-relative representation, which is less often discussed in prior works because it has not demonstrated any clear advantage. To the best of our knowledge, there are three works using the pairwise-relative representation: HDGT [28], GoRela [11] and HiVT [70]. HDGT and GoRela are based on Graph Neural Networks (GNNs), and their implementation requires message passing and GNN libraries. However, these libraries are often less efficiently implemented on Graphics Processing Units (GPUs) when compared to the basic matrix operations that Transformers rely on. HiVT augments the agent-centric encoders with a pairwise-relative interaction decoder in order to realize multi-agent prediction. In contrast to HiVT which uses agent-centric vectors and the standard Transformer, we formulate all inputs as pairwise-relative polylines and introduce the KNARPE attention mechanism. As a result, our HPTR demonstrates clear advantages in terms of accuracy and efficiency. Replacing the vanilla attention with our KNARPE, we can borrow ideas from other Transformer-based methods and adapt them to the pairwise-relative representation. For example, the hierarchical architecture of our HPTR is inspired by Wayformer [36]. More sophisticated architectures and techniques, such as goal-based decoding [21, 69], can also be incorporated into our framework to further boost the performance.

**Rasterized representation** was widely used in early works on motion prediction [34, 40, 45, 51]. These methods use convolutional neural networks (CNNs) to process the rasterized image of agent-centric ROI. To improve the inference efficiency, some works [5, 6, 10, 65] pre-compute the static map features and use rotated ROI alignment [26] to retrieve the local contexts around the target agent. In this paper, our KNARPE realizes this operation for pure Transformer-based architectures.

**Transformers** with attention mechanism [53] have achieved great success in natural language processing [13, 43] and computer vision tasks [4, 14, 64]. Inspired by the vision Transformers, we treat polylines as high-dimensional pixels; the local attribute corresponds to color channels, whereas the global pose corresponds to the pixel-wise location. We further extend the relative position encoding [12, 46] to higher dimensions and rename it to relative pose encoding (RPE) in this paper. Although the benefit of RPE is still controversial for other tasks [54, 61], its value for motion prediction is demonstrated in this paper. Instead of using RPE, previous motion prediction Transformers use everything as input, which is equivalent to concatenating the pixel-wise location to the RGB channels. This is a very uncommon practice from the perspective of vision Transformers.

## 3 Method

In Sec. 3.1 we introduce the pairwise-relative polyline representation which enjoys the advantages of both the agent-centric and the scene-centric representation. Then in Sec. 3.2 we present the **K**-nearest **N**eighbor **A**ttention with **R**elative **P**ose **E**ncoding (KNARPE) which enables the pairwise-

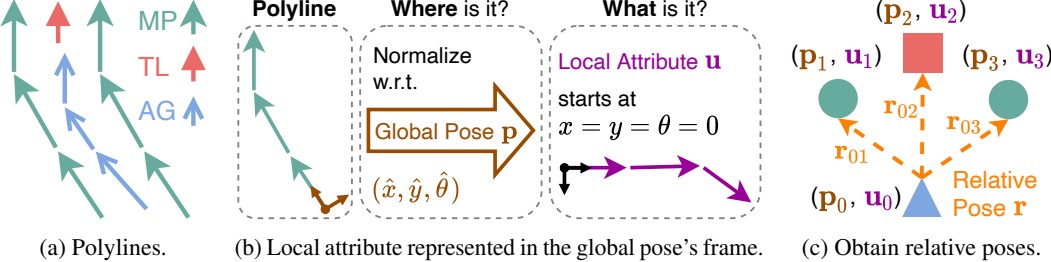

| (a) Polylines. | (b) Local attribute represented in the global pose's frame. | (c) Obtain relative poses. |

Figure 2: Pairwise-relative polyline representation. MP: Map. TL: Traffic lights. AG: Agents.

relative polyline representation to be used by Transformers. Based on KNARPE, we propose the **H**eterogeneous **P**olyline **T**ransformer with **R**elative pose encoding (HPTR) in Sec. 3.3. HPTR uses a hierarchical architecture to prevent redundant computations and to realize the asynchronous update of heterogeneous tokens. Finally in Sec. 3.4 we discuss our output representation and training strategies.

## 3.1 Pairwise-relative polyline representation

Based on prior works [11, 28], we formulate the pairwise-relative polyline representation as the third type of input representation for motion prediction. As shown in Figure 2a, all data relevant to motion prediction can be represented as an ordered list of consecutive vectors, i.e. polylines. By transforming the polyline to the coordinate frame of its global pose as done in Figure 2b, a polyline is fully described by a pair of global pose and local attribute. The global pose specifies the location and heading of the polyline in the global coordinate system, whereas the local attribute describes the polyline in its local coordinates, for example it is a yellow solid lane, 8 meters long, slightly curved to the right. In practice, it does not matter where the lane is on the earth; we only care about where the lane is relative to us. Hence, we obtain the relative poses from the global poses as shown in Figure 2c, before feeding them to the prediction network. Specifically, the global pose and local attribute of polyline $i$ are denoted as $(\mathbf{p}_i, \mathbf{u}_i)$. The global pose $\mathbf{p}_i$ has 3 degrees of freedom $(x, y, \theta)$, i.e. the 2D position and the heading yaw angle. The local attribute $\mathbf{u}_i$ is derived from $\mathbf{c}_i$, the intrinsic characteristics of the polyline, and $\mathbf{l}_i$, the polyline vectors represented in the local coordinate.

The task of marginal motion prediction is to predict the future 2D positions individually for each target agent based on the static HD map (MP), the history trajectories of all agents (AG) and the traffic lights (TL). For each scenario to be predicted, we consider a maximum number of $N_{\mathrm{MP}}$ map polylines, $N_{\mathrm{TL}}$ traffic lights and $N_{\mathrm{AG}}$ agents. We define $t = 0$ to be the current step, $\{T_{\mathrm{h}} - 1, \ldots, 0\}$ to be the observed history steps and $\{1, \ldots, T_{\mathrm{f}}\}$ to be the predicted future steps. The static HD map are spatial polylines $(\mathbf{p}_i^{\mathrm{MP}}, \mathbf{l}_i^{\mathrm{MP}}, \mathbf{c}_i^{\mathrm{MP}}), i \in \{1, \ldots, N_{\mathrm{MP}}\}$ where $\mathbf{p}_i^{\mathrm{MP}} \in \mathbb{R}^3$ is its starting vector, $\mathbf{l}_i^{\mathrm{MP}} \in \mathbb{R}^{N_{\mathrm{node}} \times 4}$ are the 2D positions and directions of the $N_{\mathrm{node}}$ segments normalized against $\mathbf{p}_i^{\mathrm{MP}}$, and $\mathbf{c}_i^{\mathrm{MP}} \in \mathbb{R}^{C_{\mathrm{MP}}}$ is the one-hot encoding of $C_{\mathrm{MP}}$ different lane types. Polygonal elements, such as crosswalks, are converted to a group of parallel polylines across the polygon. Similar to the map, history trajectories of agents are represented as spatial-temporal polylines $(\mathbf{p}_i^{\mathrm{AG}}, \mathbf{l}_i^{\mathrm{AG}}, \mathbf{c}_i^{\mathrm{AG}}), i \in \{1, \ldots, N_{\mathrm{AG}}\}$ where $\mathbf{p}_i^{\mathrm{AG}} \in \mathbb{R}^3$ is the last observed agent pose, $\mathbf{l}_i^{\mathrm{AG}} \in \mathbb{R}^{T_{\mathrm{h}} \times (6+3)}$ contains the history 2D positions, directions and velocities normalized against $\mathbf{p}_i^{\mathrm{AG}}$ as well as the 1D speed, yaw rate and acceleration which do not need to be normalized, and $\mathbf{c}_i^{\mathrm{AG}} \in \mathbb{R}^6$ is the 3D agent size and the one-hot encoding of 3 agent types (vehicle, pedestrian and cyclist). Following VectorNet [17], we use PointNet [41] with masked max-pooling to aggregate $(\mathbf{l}_i^{\mathrm{MP}}, \mathbf{c}_i^{\mathrm{MP}})$ into $\mathbf{u}_i^{\mathrm{MP}} \in \mathbb{R}^D$ and $(\mathbf{l}_i^{\mathrm{AG}}, \mathbf{c}_i^{\mathrm{AG}})$ into $\mathbf{u}_i^{\mathrm{AG}} \in \mathbb{R}^D$, where $D$ is the hidden dimension. Since current datasets contain only the detection results rather than the tracking results of traffic lights, we consider only the traffic lights observed at $t = 0$ and represent them as singular polylines $(\mathbf{p}_i^{\mathrm{TL}}, \mathbf{c}_i^{\mathrm{TL}}), i \in \{1, \ldots, N_{\mathrm{TL}}\}$ where $\mathbf{p}_i^{\mathrm{TL}} \in \mathbb{R}^3$ is the pose of the stop point and $\mathbf{c}_i^{\mathrm{TL}} \in \mathbb{R}^{C_{\mathrm{TL}}}$ is the one-hot encoding of $C_{\mathrm{TL}}$ different states of traffic lights. We use a multi-layer perceptron (MLP) to encode $\mathbf{c}_i^{\mathrm{TL}}$ into $\mathbf{u}_i^{\mathrm{TL}} \in \mathbb{R}^D$.

Because the local attributes are normalized and the global poses are used to compute the relative poses before being consumed by the network, the pairwise-relative representation preserves the viewpoint invariance of the agent-centric representation. Additionally, since the local attributes have higher dimensions compared to the 3-dimensional global poses, sharing the local attributes enables the pairwise-relative representation to maintain the good scalability of the scene-centric representation.

However, so far this representation has only be exploited by GNNs, which are not comparable to Transformers in terms of accuracy and efficiency on the motion prediction benchmarks [1, 58].

## 3.2 KNARPE: K-nearest neighbors attention with relative pose encoding

After encoding the local attributes via the polyline-level encoders, the scenario is now described as $(\mathbf{p}_i, \mathbf{u}_i), i \in \{1, \ldots, N\}$ where $N = N_{\text{MP}} + N_{\text{AG}} + N_{\text{TL}}$ is the total number of polylines. However, this representation cannot be handled by standard self-attention because on the one hand modeling the all-to-all attention is prohibitively expensive [36] due to the large $N$, and on the other hand the performance deteriorates when the input is scene-centric [37]. To address both problems, we introduce the **K**-nearest **N**eighbors **A**ttention with **R**elative **P**ose **E**ncoding (KNARPE). Similar to MTR [47], KNARPE limits the attention to the K-nearest neighbors of each token. Specifically, $\kappa_i^K(d_{i1}, \ldots, d_{iN}) \subseteq \{1, \ldots, N\}$ is a set that contains the indices of $K$ tokens closest to token $i$. For simplicity, we use the L2 distance to measure the distance $d_{ij}$ between the global poses of token $i$ and $j$. Instead of directly processing global poses, KNARPE uses the relative pose encoding (RPE), i.e. the positional encoding for pairwise-relative poses. Denoting $\mathbf{r}_{ij} = (x_{ij}, y_{ij}, \theta_{ij})$ to be the global pose of token $j$ represented in the coordinate of token $i$, i.e. $\mathbf{p}_j$ transformed to the coordinate of $\mathbf{p}_i$, the RPE of $\mathbf{r}_{ij}$ is computed using sinusoidal positional encoding (PE) and angular encoding (AE) [68],

$$\text{RPE}(\mathbf{r}_{ij}) = \text{concat}(\text{PE}(x_{ij}), \text{PE}(y_{ij}), \text{AE}(\theta_{ij})), \tag{1}$$

$$\text{PE}_{2i}(x) = \sin(x \cdot \omega^{\frac{2i}{D}}), \ \text{PE}_{2i+1}(x) = \cos(x \cdot \omega^{\frac{2i}{D}}), \tag{2}$$

$$\text{AE}_{2i}(\theta) = \sin\left(\theta \cdot (i+1)\right), \ \text{AE}_{2i+1}(\theta) = \cos\left(\theta \cdot (i+1)\right), i \in \{0, \ldots, D/2 - 1\}, \tag{3}$$

where $\omega$ is the base frequency. Following [46], the RPE is projected and added to the keys and values to obtain $\mathbf{z}_i$, the output of letting token $i$ attend to its $K$ neighbors $\kappa_i^K$,

$$\mathbf{z}_i = \text{KNARPE}\left(\mathbf{u}_i, \mathbf{u}_j, \mathbf{r}_{ij} \mid j \in \kappa_i^K\right) = \sum_{j \in \kappa_i^K} \alpha_{ij}\left(\mathbf{u}_j\mathbf{W}^v + \mathbf{b}^v + \text{RPE}(\mathbf{r}_{ij})\hat{\mathbf{W}}^v + \hat{\mathbf{b}}^v\right), \tag{4}$$

$$\alpha_{ij} = \frac{\exp(e_{ij})}{\sum_{k \in \kappa_i^K} \exp(e_{ik})}, \quad e_{ij} = \frac{(\mathbf{u}_i\mathbf{W}^q + \mathbf{b}^q)(\mathbf{u}_j\mathbf{W}^k + \mathbf{b}^k + \text{RPE}(\mathbf{r}_{ij})\hat{\mathbf{W}}^k + \hat{\mathbf{b}}^k)}{\sqrt{D}}, \tag{5}$$

where $\alpha_{ij}$ are the attention weights, $e_{ij}$ are the logits, $W^{\{q,k,v\}}, b^{\{q,k,v\}}$ are the learnable projection matrices and biases for query, key and value, and $\hat{W}^{\{k,v\}}, \hat{b}^{\{k,v\}}$ are the learnable projection matrices and biases for RPE. We do not apply RPE to query because doing this does not boost the performance in our experiments.

Efficient implementation of KNARPE can be achieved using basic matrix operations such as matrix indexing, summation and element-wise multiplication. See the appendix for more details. KNARPE allows the pairwise-relative representation to be used by pure Transformer-based architectures. Self-attention with KNARPE aggregates the local context for each token in the same way as CNN aggregates the context around each pixel via convolutional kernels. The pixel corresponds to the token, whereas the kernel size of a CNN corresponds to the number of neighbors of KNARPE. Cross-attention with KNARPE enables rotated ROI alignment of pre-computed features, e.g. the static map features. Previously, this was only possible for CNN-based [6, 10] and GNN-based [11, 28] methods.

## 3.3 HPTR: Heterogeneous polyline transformer with relative pose encoding

By replacing the standard multi-head attention [53] with our KNARPE, we construct Transformer encoders and decoders which can model the interactions between heterogeneous polylines via relative poses and local attributes. To address the marginal motion prediction task involving HD map, traffic lights and agents, we propose the **H**eterogeneous **P**olyline **T**ransformer with **R**elative pose encoding (HPTR) as shown in Figure 3. Since the temporal dimension is eliminated by the polyline-level encoder [17], HPTR models only the spatial relationship between tokens from different classes.

Firstly in Figure 3a, the intra-class Transformer encoders build a block diagonal attention matrix that models the interactions within each class of tokens. Then in Figure 3b, the inter-class Transformer decoders enhance the traffic lights tokens by making them attend to the map tokens, whereas the agent tokens are enhanced by attending to both the traffic lights and map tokens. The intuition behind this is the hierarchical nature of traffic; first there is only the map, then the traffic lights are added

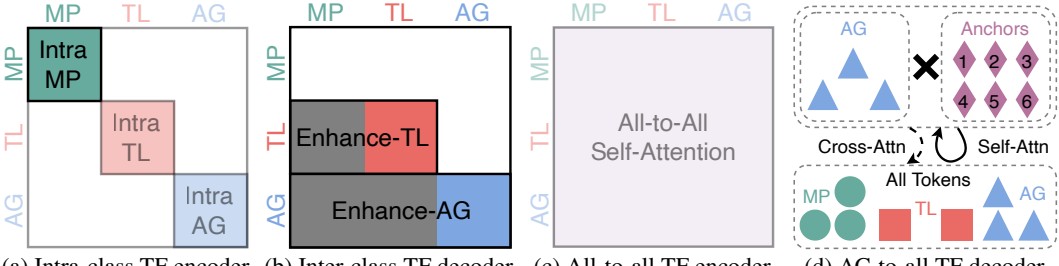

(a) Intra-class TF encoder  (b) Inter-class TF decoder  (c) All-to-all TF encoder  (d) AC-to-all TF decoder

Figure 3: The hierarchical architecture of HPTR. Transformers are applied in sequential order from left to right, top to down. Some attentions are redundant and can be skipped for better efficiency. We propose the lower triangular attention matrix, which excludes the intra-TL, intra-AG and all-to-all self-attentions (the transparent parts). By removing the redundant attentions, this specific lower triangular architecture enables asynchronous token update during online inference. TF: Transformer. Attn: Attention. MP: Map. TL: Traffic lights. AG: Agents.

and finally the agents join. Intuitively, the map influences the interpretation of traffic lights but not vice versa, whereas map and traffic lights together influence the behavior of agents but not vice versa. After the inter-class Transformer decoders, the all-to-all Transformer encoder in Figure 3c builds a full attention matrix allowing tokens to attend to each other irrespective of their class. Finally, each agent token is concatenated with $N_{AC}$ learnable anchors, which are shared among each type of agent. Since the task is marginal motion prediction, we batch over agents and anchors, i.e. now the number of anchor tokens is $N_{AG} \cdot N_{AC}$. Then the anchor-to-all Transformer decoder in Figure 3d lets each of these anchor tokens attend to all tokens so as to aggregate more contextual information. The final output is denoted as $\hat{\mathbf{z}}_i^{AG}, i \in \{1, \ldots, N_{AG} \cdot N_{AC}\}$, and it is used to generate the multi-modal future trajectories. We set the number of neighbors to $K$ for the intra-class and all-to-all Transformers. This number $K$ is multiplied by $\gamma_{TL}, \gamma_{AG}$ and $\gamma_{AC}$ respectively for the enhance-TL, enhance-AG and AC-to-all Transformers, such that these Transformers can have a larger receptive field.

HPTR organizes the Transformers in a hierarchical way such that some of the intermediate results can be cached and reused during the online inference; for example the outputs of the intra-MP Transformer, i.e. the static map features. This allows tokens from different classes to be updated asynchronously, which significantly reduces the online inference latency. We can further improve the efficiency without sacrificing the performance by trimming the redundant attentions. The attention matrices shown in Figure 3a, 3b and 3c are overlapping, which means some relationships are repeatedly modeled, such as the agent-to-agent self-attention. We propose to remove the intra-TL, intra-AG and all-to-all Transformer, while keeping the intra-MP, enhance-TL and enhance-AG Transformer. These three Transformers together build a lower triangular attention matrix which is necessary and sufficient to model the relationship between map, traffic lights and agents. Our approach can be seen as an extension to Wayformer [36] which does not introduce the inter-class Transformer decoders. We can trim Figure 3 differently and cast HPTR into the Wayformer. Specifically, Wayformer with late fusion corresponds to HPTR with diagonal attention matrix (only intra-class TF), Wayformer with early fusion corresponds to HPTR with full attention matrix (only all-to-all TF) and Wayformer with hierarchical fusion corresponds to HPTR with the diagonal followed by the full attention matrix.

### 3.4 Output representation and training strategies

We follow the common practice [36, 47, 52] to represent the outputs as a mixture of Gaussians and train with hard assignment. The multi-modal future trajectories for each agent are generated by decoding $\hat{\mathbf{z}}_i^{AG}, i \in \{1, \ldots, N_{AG} \cdot N_{AC}\}$ via two MLPs; the confidence head and the trajectory head. The confidence head predicts a scalar confidence of each trajectory. Besides the Gaussian parameters of 2D positions $(\mu_x, \mu_y, \sigma_x, \sigma_y, \rho)$ at each future time step, our trajectory head predicts also the yaw angles, speeds and 2D velocities. We use the cross entropy loss for confidences, negative log-likelihood loss for 2D positions, negative cosine loss for yaw angles and Huber loss for speeds and 2D velocities. The final training loss is the unweighted sum of all losses. Following the hard-assignment strategy, for each agent we optimize only the predicted trajectory that is closest to the ground truth in terms of 2D average displacement error. Please refer to the appendix for more details.

# 4 Experiments

## 4.1 Experimental setup

**Benchmarks.** We benchmark our method on the two most popular datasets: the Waymo Open Motion Dataset (WOMD) [16] and the Argoverse-2 motion forecasting dataset (AV2) [60]. Both datasets have an online leaderboard for marginal motion prediction. However, their task descriptions are slightly different. For each scenario, WOMD evaluates the predictions of up to 8 agents, whereas AV2 evaluates only one agent. Both datasets require exactly 6 futures to be predicted for each target agent. The sampling time is 0.1 seconds for both datasets. The history length is 11 steps for WOMD and 50 steps for AV2, whereas the future length is 80 steps for WOMD and 60 steps for AV2. We use the official evaluation tool of the leaderboards to compute the metrics. For the WOMD leaderboard [58], the ranking metric is soft mAP; for the AV2 leaderboard [1], it is brier-minFDE. Please refer to the leaderboard homepages for more details about the dataset and evaluation metrics.

**Implementation details.** We use Transformer with pre-layer normalization [62] for HPTR. For each episode, we consider $N_{MP} = 1024$ map polylines, $N_{TL} = 40$ traffic light stop points and $N_{AG} = 64$ agents. Each polyline contains up to $N_{node} = 20$ one-meter-long segments. The base number of neighbors considered by KNARPE is $K = 36$. This number is multiplied by $\gamma_{TL} = 2$, $\gamma_{AG} = 4$ and $\gamma_{AC} = 10$ respectively for the enhance-TL, enhance-AG and AC-to-all Transformer. We set $N_{AC} = 6$ to predict exactly 6 futures as specified by the leaderboard. We do not apply ensembling or expensive post-processing such as trajectory aggregation. Our post-processing manipulates only the confidences. To improve the soft mAP, we use the non-maximum suppression of MPA [31] for the WOMD leaderboard. To improve the brier-minFDE, we set the softmax temperature to 0.5 for the AV2 leaderboard. More details are provided in the appendix.

**Training details.** Thanks to the viewpoint invariance of the pairwise-relative representation, no data augmentation or input permutation is needed for the training of HPTR. We use AdamW optimizer with an initial learning rate of 1e-4 and decaying by 0.5 every 25 epochs. We train with a total batch size of 12 episodes on 4 RTX 2080Ti GPUs. For WOMD, we randomly sample 25% from all training episodes at each epoch; for AV2 we use 50%. Our final models are trained for 120 epochs for WOMD and 150 epochs for AV2. The complete training takes 10 days. For WOMD, the SDV agents, agents of interest and agents to be predicted are used for optimization. For AV2, the SDV agents, scored agents and focal agents are used for optimization. To address the imbalance between agent types, for WOMD we additionally optimize for pedestrians and cyclists which are tracked for at least 4 seconds.

## 4.2 Benchmark results

In Table 1 we benchmark our approach on the public leaderboards of WOMD and AV2. From the leaderboard we observe that using ensemble and predicting redundant trajectories can increase the prediction diversity and hence improve the soft mAP, brier-minFDE and miss rate. For example, MTR-Adv-ens aggregates the outputs of 7 ablation models, each predicting 64 futures. Besides the exaggerated large number of redundant predictions, it is also non-trivial to aggregate them into 6 predictions. Some works use K-means clustering while the others use non-maximum suppression; both involve heuristic parameter fine-tuning. Since our framework is dedicated to real-world autonomous driving, applying these computationally expensive techniques is contradictory to our motivation. For a fair comparison, we focus on end-to-end methods which do not apply these techniques. On the WOMD dataset, we achieve SOTA performance among the end-to-end methods, including MTR-e2e, the end-to-end version of MTR, and MPA, the end-to-end version of MultiPath++. Specifically, HPTR outperforms the pairwise-relative HDGT and the scene-centric SceneTransformer by a large margin in all metrics. We also show competitive performance on the AV2 leaderboard. We achieve better performance in all metrics except the brier-minFDE compared to GoRela, which uses GNNs to tackle the pairwise-relative representation. Since there exists a performance gap while adapting from one dataset to another and we choose WOMD to be our main benchmark, our performance on the AV2 leaderboard could be further improved by fine-tuning the hyper-parameters for the AV2 dataset.

## 4.3 Ablation study

In Table 2 we ablate different input representations and hierarchical architectures. The scene-centric baseline, HPTR SC, uses the architecture of our HPTR and the input representation of

Table 1: Results on the marginal motion prediction leaderboards of WOMD and AV2. Both tables are sorted according to the ranking metric such that the best performing method is on the top. The ranking metric is *soft mAP* for WOMD, and *brier-minFDE$_6$* for AV2. † denotes ensemble. * denotes predicting more futures than required. SC: scene-centric. AC: agent-centric. PR: pairwise-relative.

| WOMD *test* | repr. | *soft mAP* ↑ | mAP ↑ | minADE ↓ | minFDE ↓ | miss rate ↓ |
|---|---|---|---|---|---|---|
| *†MTR-Adv-ens [47] | AC | 0.4594 | 0.4492 | 0.5640 | 1.1344 | 0.1160 |
| *†Wayformer [36] | AC | 0.4335 | 0.4190 | 0.5454 | 1.1280 | 0.1228 |
| *MTR [47] | AC | 0.4216 | 0.4129 | 0.6050 | 1.2207 | 0.1351 |
| *†MultiPath++ [52] | AC | N/A | 0.4092 | 0.5557 | 1.1577 | 0.1340 |
| **HPTR (Ours)** | PR | 0.3968 | 0.3904 | 0.5565 | 1.1393 | 0.1434 |
| MPA [31] (MultiPath++) | AC | 0.3930 | 0.3866 | 0.5913 | 1.2507 | 0.1603 |
| HDGT [28] | PR | 0.3709 | 0.3577 | 0.7676 | 1.1077 | 0.1325 |
| Gnet [18] | AC | 0.3396 | 0.3259 | 0.6207 | 1.2391 | 0.1718 |
| SceneTransformer [37] | SC | N/A | 0.2788 | 0.6117 | 1.2116 | 0.1564 |

| WOMD *valid* | repr. | *soft mAP* ↑ | mAP ↑ | minADE ↓ | minFDE ↓ | miss rate ↓ |
|---|---|---|---|---|---|---|
| **HPTR (Ours)** | PR | 0.4222 | 0.4150 | 0.5378 | 1.0923 | 0.1326 |
| MTR-e2e | AC | N/A | 0.3245 | 0.5160 | 1.0404 | 0.1234 |

| AV2 *test* | repr. | *brier-minFDE$_6$* ↓ | minFDE$_6$ ↓ | minFDE$_1$ ↓ | minADE$_6$ ↓ | miss rate$_6$ ↓ |
|---|---|---|---|---|---|---|
| *ProphNet [55] | AC | 1.88 | 1.33 | 4.74 | 0.68 | 0.18 |
| †Gnet [18] | AC | 1.90 | 1.34 | 4.40 | 0.69 | 0.18 |
| †TENET [56] | AC | 1.90 | 1.38 | 4.69 | 0.70 | 0.19 |
| *MTR [47] | AC | 1.98 | 1.44 | 4.39 | 0.73 | 0.15 |
| GoRela [11] | PR | 2.01 | 1.48 | 4.62 | 0.76 | 0.22 |
| **HPTR (Ours)** | PR | 2.03 | 1.43 | 4.61 | 0.73 | 0.19 |
| THOMAS [19] | AC | 2.16 | 1.51 | 4.71 | 0.88 | 0.20 |
| HDGT [28] | PR | 2.24 | 1.60 | 5.37 | 0.84 | 0.21 |

SceneTransformer [37]. The agent-centric baseline, WF baseline, is our reimplementation of the Wayformer [36] with multi-axis attention and early fusion. Both baselines achieve their expected performances compared to their original implementations. The large performance gap between HPTR SC and other models confirms the disadvantage of scene-centric representation. The agent-centric baseline requires more training iterations. After convergence, its performance is on par with our HPTR. To ablate the hierarchical architecture, we implement three variations of HPTR; each corresponds to a fusion strategy investigated by Wayformer. The full attention corresponds to the early fusion, diagonal corresponds to late fusion, and HPTR with diagonal followed by full attention corresponds to Wayformer with hierarchical fusion. While the early fusion performs the best for Wayformer, for HPTR the lower triangular attention we proposed outperforms both the early and the late fusion by a significant margin. The performance of hierarchical fusion is slightly worse than ours, but its inference latency is significantly longer because of the redundancy in its attention matrices.

## 4.4 Efficiency analysis and qualitative results

In Figure 4 we compare the computational efficiency of the ablation models presented in Table 2. As discussed in prior works [36, 47], one of the major drawbacks of agent-centric approaches is the poor scalability, which is reflected by the large slope of the memory and latency curves of our WF baseline. On the RTX 2080Ti, it can handle only up to 48 agents while the inference latency is 140ms. On the contrary, the scene-centric baseline is extremely efficient, but it suffers from poor accuracy. Our HPTR takes the best of both approaches. In terms of efficiency, our HPTR is comparable to the scene-centric approaches, while our performance is on par with the agent-centric approaches. Compared to the scene-centric baseline, the GPU memory consumption and the inference latency of HPTR are slightly higher because for each new agent, we have to compute its relative pose to all existing tokens. Compared to the hierarchical architectures introduced by Wayformer, HPTR with our lower triangular attention achieves the best trade-off between accuracy and latency. By reusing the static map features during the online inference, our HPTR predicts the multi-modal futures for 64 agents in 37ms without the use of inference libraries. Narrowing the receptive field, for example by

Table 2: Ablation on the valid split of WOMD. The table is sorted according to the *soft mAP* such that the best-performing method is on the top. Performances are reported as the mean plus-minus 3 standard deviations over 3 training seeds. Models are trained for 60 epochs if not otherwise mentioned. WF: Wayformer. SC: scene-centric, AC: agent-centric, PR: pairwise-relative.

| WOMD *valid* | input repr. | intra MP | intra TL/AG | enhance TL/AG | all 2all | minFDE ↓ | *soft mAP* ↑ |
|---|---|---|---|---|---|---|---|
| **HPTR (Ours)** | PR | ✓ | × | ✓ | × | $1.145 \pm 0.016$ | $0.399 \pm 0.010$ |
| WF baseline (100-epoch) | AC | × | × | × | ✓ | $1.161 \pm 0.006$ | $0.397 \pm 0.007$ |
| HPTR diag+full (WF hier.) | PR | ✓ | ✓ | × | ✓ | $1.156 \pm 0.014$ | $0.391 \pm 0.002$ |
| HPTR diag (WF late) | PR | ✓ | ✓ | × | × | $1.169 \pm 0.013$ | $0.387 \pm 0.012$ |
| HPTR full (WF early) | PR | × | × | × | ✓ | $1.158 \pm 0.093$ | $0.386 \pm 0.041$ |
| WF baseline | AC | × | × | × | ✓ | $1.212 \pm 0.019$ | $0.378 \pm 0.014$ |
| HPTR SC | SC | ✓ | × | ✓ | × | $1.687 \pm 0.046$ | $0.246 \pm 0.005$ |

Figure 4: HPTR is as efficient as scene-centric methods in terms of GPU memory consumption and inference latency, while being as accurate as agent-centric methods. We use standard Ubuntu, Python and Pytorch without optimizing for real-time deployment. We predict one scenario at each inference time on one 2080Ti, i.e. we batch over scenarios and the batch size is 1. The number of context agents is the same as the number of predicted agents for all experiments. During offline inference, every inference starts from scratch, while during online inference, we cache and reuse the static map features. Specifically, we repeat the inference of the same scenario for 100 times to simulate online inference, where the static map features are computed at the first step and reused for the next 99 steps.

reducing $\lambda_{AC}$ from 10 to 8, can improve the inference speed but the accuracy might be affected in some cases. Using half precision at inference time can reduce the latency to 25ms (40 fps) without affecting the accuracy. Compared to the most efficient agent-centric method Wayformer, we reduce the memory consumption and the online inference latency by 80% without sacrificing the accuracy.

In Figure 5 we compare the computational efficiency of our method with the GNN-based pairwise-relative methods. Currently there are two such methods, GoRela [11] and HDGT [28]. While HDGT does not match the prediction accuracy of GoRela or our method, it is worth noting that GoRela is not open-sourced, whereas HDGT is. Therefore, we use HDGT in this efficiency comparison. The left plot of Figure 5 confirms the good scalability of HDGT in terms of GPU memory. This is expected because it uses the pairwise-relative representation. In the middle plot, we can observe that HDGT is slower than both our HPTR and our agent-centric Wayformer baseline in terms of offline inference speed. To confirm that HDGT runs correctly on our setup, in the right plot we reproduce the inference time of HDGT on the complete WOMD validation split with different validation batch size and we compare the reproduce numbers with the reported number in the HDGT paper. The slow inference speed of GNN-based methods such as HDGT is mainly because GNN libraries cannot utilize the GPU as efficiently as the basic matrix operations do. Our KNARPE is implemented with the most basic matrix operations, hence it is better suited for real-time and on-board applications.

Figure 6 illustrates the qualitative results of our HPTR. For each type of agent, we select a successful case where the most confident prediction matches the ground truth, and a failure case to show the limitation of our method. In the successful cases, we observe the vehicle stops at the red light, the pedestrian walks along the road edge with another person, and the cyclist rides across the road via the

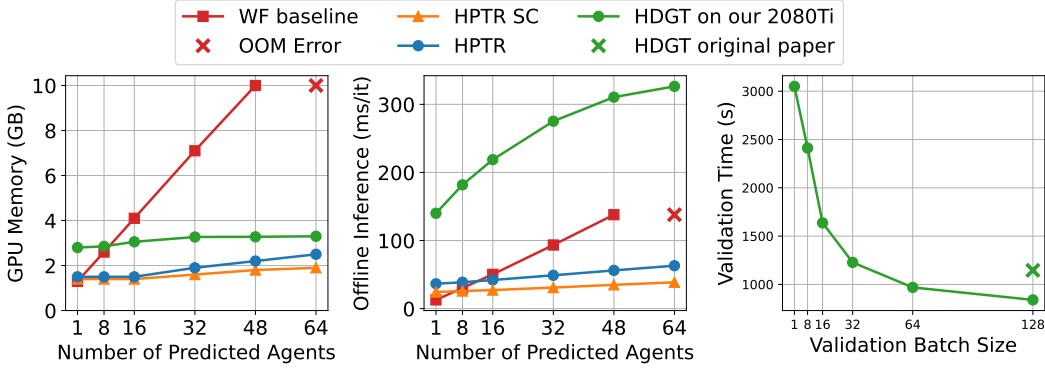

Figure 5: Efficiency comparison with HDGT. We run their official repository on our machine with a single 2080Ti. The left plot shows that HDGT achieves good scalability in terms of GPU memory consumption as expected. The middle plot shows that the offline inference latency (with batch size 1) of HDGT scales well, but it is significantly larger than that of other Transformer-based methods. The right plot shows the inference times for the complete WOMD validation split (64 agents per episode) with different validation batch sizes. It confirms the inference speed reported in the original HDGT paper is correctly reproduced on our setup. Our setup is faster because it has a more powerful CPU.

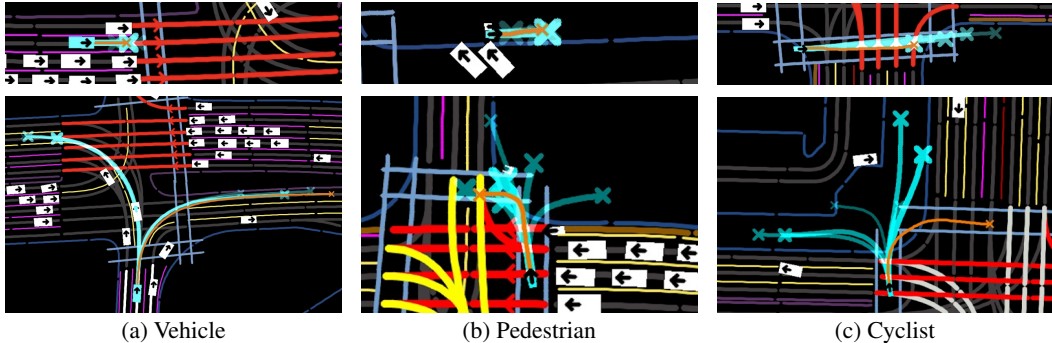

| (a) Vehicle | (b) Pedestrian | (c) Cyclist |

Figure 6: Qualitative results of HPTR. For each type of agent we show a successful case (top) and a failure case (bottom). The intersections are cluttered because we visualize traffic lights by overlaying the lanes they control with their color. The ground truth is in orange. The target agent and the predictions are in cyan. The most confident prediction has the least transparent color, the thickest line and the biggest cross. Please refer to the appendix for a detailed explanation of the visualization.

crosswalk. Although in the failure cases the most confident prediction deviates from the ground truth, we are encouraged to see that our predictions are still reasonable. The failure cases can be addressed by improving the prediction diversity via goal-conditioning, which is orthogonal to our contributions.

## 5 Conclusion

In this paper we introduce a novel attention module, KNARPE, that allows the pairwise-relative representation to be used by Transformers. Based on KNARPE, we present a pure Transformer-based framework called HPTR, which uses hierarchical architecture to enable asynchronous token update and avoid redundant computations. While agent-centric methods suffer from poor scalability and scene-centric methods suffer from poor accuracy, our HPTR gets the best from both worlds. Experiments on the two most popular benchmarks show that our approach achieves superior performance, while satisfying the real-time and on-board requirements of real-world autonomous driving.

**Limitations.** The poses in this work reside on a 2D plane, which can be extended to the 3D space in the future. For simplicity, we use L2 distance to obtain the K-nearest neighbors. Future works can explore more sophisticated distances which involve map topology. Currently we use the most basic anchor-based decoder which has limited diversity. This can be improved by using more advanced decoding techniques, such as goal-conditioning. In this paper we focus on marginal motion prediction. It would be interesting to investigate the potential of our approach in other prediction tasks.

## Acknowledgments and Disclosure of Funding

This work is funded by Toyota Motor Europe via the research project TRACE-Zürich.

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

# A  Output representation and training strategies

For each anchor token $\hat{\mathbf{z}}_i^{\text{AG}}, i \in \{1, \ldots, N_{\text{AG}} \cdot N_{\text{AC}}\}$, the confidence head predicts the logits $p_k, k \in \{1, \ldots, 6\}$, whereas the trajectory head predicts 6 trajectories, each of which is represented as $(\mu_x^t, \mu_y^t, \log \sigma_x^t, \log \sigma_y^t, \rho^t, v_x^t, v_y^t, \theta^t, s^t), t \in \{1, \ldots, T_f\}$, i.e. the mean of the Gaussian in $x, y$, the log standard deviation of the Gaussian in $x, y$, the correlation of the Gaussians, the velocity in $x, y$, the heading angle and the speed. We denote the ground truth as $(\hat{x}, \hat{y}, \hat{v}_x, \hat{v}_y, \hat{\theta}, \hat{s})$. The negative log-likelihood loss for position is formulated as

$$L_{\text{pos}} = -\log \mathcal{N}(\hat{x}, \hat{y} \mid \mu_x, \mu_y, \sigma_x, \sigma_y, \rho). \tag{6}$$

The negative cosine loss for the heading angle is formulated as

$$L_{\text{rot}} = -\cos(\hat{\theta} - \theta). \tag{7}$$

The Huber loss for velocities and speed is formulated as

$$L_{\text{vel}} = \mathcal{L}_\delta(\hat{v}_x - v_x) + \mathcal{L}_\delta(\hat{v}_y - v_y) + \mathcal{L}_\delta(\hat{s} - s), \tag{8}$$

where $\mathcal{L}_\delta$ is the Huber loss. We use $\delta = 1$ for all Huber losses. The final regression loss for a trajectory is the unweighted sum

$$L_{\text{traj}} = L_{\text{pos}} + L_{\text{rot}} + L_{\text{vel}}, \tag{9}$$

which is averaged over the future time steps where the ground truth is available. We use a hard assignment strategy, i.e. among the 6 predictions of each agent we select the one that is closest to the ground truth in terms of average displacement error and optimize only for that prediction. Denoting the index of this prediction as $\hat{k}$, we train the confidence head via the cross entropy loss by taking $\hat{k}$ as the ground truth:

$$L_{\text{conf}} = -\log \frac{\exp(p_{\hat{k}})}{\sum_{i=1}^6 \exp(p_i)}. \tag{10}$$

The final training loss for the complete model is the unweighted sum

$$L = L_{\text{traj}} + L_{\text{conf}}. \tag{11}$$

Our output representation and training strategies are the same as prior works [36, 47, 52], except for the auxiliary losses on velocities, speeds and heading angles.

# B  Implementation details

## B.1  KNARPE implementation

Figure 7 shows how KNARPE is implemented with the most basic matrix operations, i.e. matrix indexing, summation and element-wise multiplication.

## B.2  Network architectures

We use ReLU activation and set the hidden dimension $D$ to 256. Our KNARPE is implemented with multi-head attention with 4 heads. We use Transformer with pre-layer normalization [62] with a dropout rate of 0.1. The feed-forward hidden dimension of Transformers is set to 1024. The base frequency $\omega$ of the sinusoidal positional encoding is set to 1000. We train a single model for all types of agents, while each type of agent has its own anchors. The polyline-line level PointNet and MLPs have 3 layers, the intra-MP Transformer encoder has 6 layers, and the inter-class as well as the AC-to-all Transformer decoders, have 2 layers. Our HPTR has 15.2M trainable parameters in total. The same setup is used for both the WOMD dataset and the AV2 dataset.

In the following, we report the configuration of ablation models. The HPTR with diagonal attention has 6 layers of intra-MP, 3 layers of intra-TL and 3 layers of intra-AG Transformer. It has 15.4M trainable parameters. The HPTR with full attention has 6 layers of all-to-all and 6 layers of AC-to-all Transformer. It has 15.2M parameters. The HPTR with diagonal followed by full attention has 6 layers of intra-MP, 2 layers of intra-TL, 2 layers of intra-AG, 2 layers of all-to-all and 2 layers of AC-to-all Transformer. It has 15.4M trainable parameters. Both the HPTR with full attention

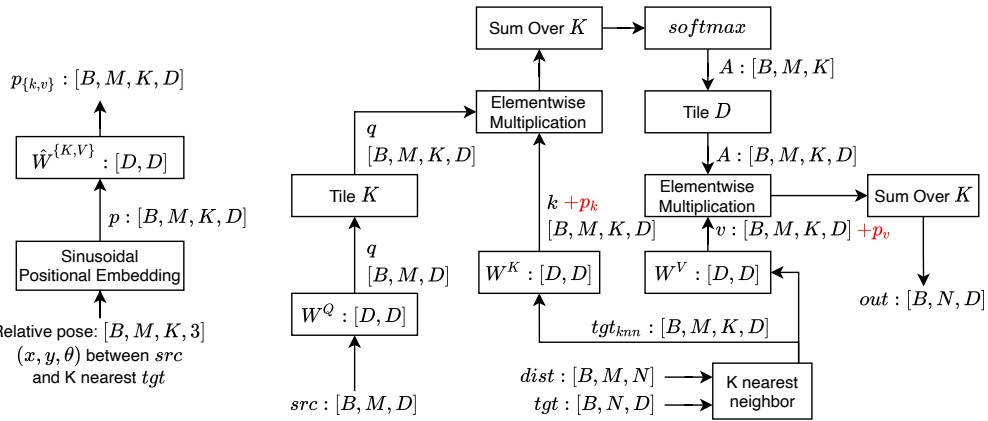

Figure 7: In contrast to most GNNs, KNARPE is implemented with the most basic matrix operations. The tensor size is shown in the brackets, where $B$ is the batch size, $M$ is the source ($src = q$) sequence length, $N$ is the target ($tgt = k = v$) sequence length, $K$ in the number of neighbors, $D$ is the hidden dimension.

and the HPTR with diagonal followed by full attention have to be trained on GPUs with 24GB of VRAM (RTX 3090 in our case) because they require more GPU memory at training time. The scene-centric baseline uses the scene-centric representation and the standard Transformer. Following SceneTransformer [37], the input 2D positions and 2D directions are pre-processed using sinusoidal positional encoding. The base frequency is set to 1000 for 2D positions and 10 for 2D directions. The output dimension of the positional encoding is 256. This model has 13M trainable parameters. The agent-centric baseline closely follows Wayformer [36]. It has 6 layers of all-to-all Transformer and 8 layers of AC-to-all Transformer. The number of latent queries is 192. The learning rate starts at 2e-4 and it is multiplied by 0.5 every 20 epochs. The training of the agent-centric baseline takes 100 epochs to converge. We do not use auxiliary losses on velocities, speeds and yaw angles to train this model. This model has 15.6M trainable parameters.

### B.3 Pre-processing and post-processing

Our pre-processing and post-processing closely follow MPA [31]. The post-processing manipulates only the confidences via greedy non-maximum suppression. The distance threshold is 2.5m for vehicles, 1m for pedestrians and 1.5m for cyclists. We use the average displacement error to compute the distance between predicted trajectories. For AV2 we simply use softmax with a temperature of 0.5 instead of doing non-maximum suppression.

### B.4 Training details

Due to the large size of motion prediction datasets, each epoch would take a very long time if trained on the complete training split. In order to track losses more frequently, we randomly sample a fraction of all training data in each epoch. This is equivalent to using the complete training dataset if the training runs for many epochs. We observe a statistically significant correlation between the model performance and the initialization of anchors. We recommend to use a large variance for the initialization distributions. Specifically, we use Xavier initialization and multiply the initialized values by 5.

Our final models for the leaderboard submission are trained for 10 days and models for ablation and development are trained for 5 days. This long training time is because on the one hand WOMD is a very large-scale dataset, and on the other hand we only use 4 RTX 2080Ti GPUs for the training. While comparing the wall time duration of training, the computational resources should be taken into consideration. As a reference, HDGT [28] uses 8 V100 and trains for 4-5 days, GoRela [11] uses 16 GPUs (model not specified but most likely A100/V100), MTR [47] uses 8 RTX 8000, Wayformer [36] uses 16 TPU v3 cores and ProphNet [55] uses 16 V100. All of these methods use a much higher number of more powerful GPUs than we use. Given comparable computational resources, the training time of our method could be reduced to 1-2 days. In terms of sample efficiency, our method is on par

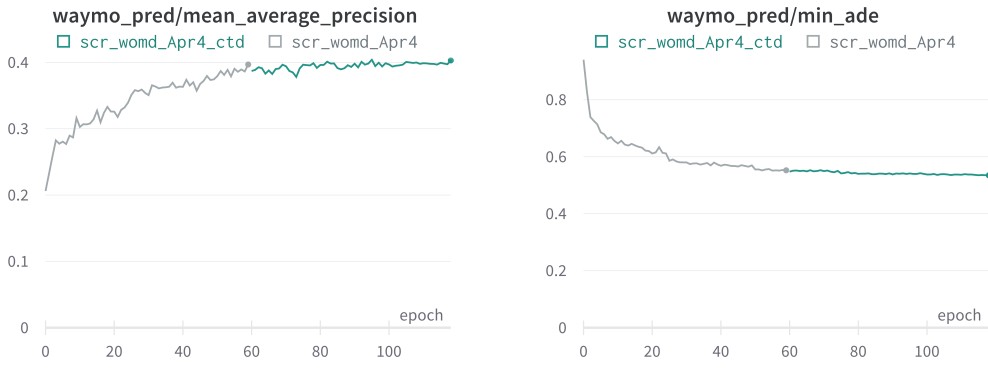

Figure 8: The validation mAP and minFDE logged during the training of our HPTR. The training runs on 4 RTX 2080Ti GPUs. The gray curve is trained for 5 days and the green curve continues the training for another 5 days. At each epoch, we sample 25% of the complete training split so as to do validation and log metrics more frequently. As a consequence, the effective epoch of these plots should be divided by 4, i.e. in the first 5 days we actually go through the complete WOMD training split 15 times. As we can see, training models for 5 days is enough for development. Only the models for the final leaderboard submission are trained for 10 days. The training can be speeded up given more computational resources.

with other methods. As shown in Figure 8, our HPTR converges after 15 epochs (5 days) and our final model is trained for 30 epochs (10 days) on WOMD. As a reference, HDGT is trained for 30 epochs, MTR is trained for 30 epochs and ProphNet is trained for 60 epochs on WOMD.

## C  Explanation of the visualization

We use white line for lane centers of freeway, aluminium line for lane centers of surface street, dark orange line for stop sign, chocolate line for lane centers of bike lane, dark blue line for road edges boundary, dark plum line for road edges median, butter line for all types of broken road lines, magenta line for all types of solid single road lines, scarlet red line for all types of double road lines, chameleon line for speed bumps and entrances to driveways, sky blue line for crosswalks.

The intersections are cluttered because we visualize traffic lights by overlaying the lanes they control with their color: red line for stop light state, yellow line for caution light state, green line for go light state, light aluminum line for unknown light state, violet line for flashing light state.

The ground truth is in orange. The target agent and the predictions are in cyan. Confidence are reflected by the transparency and thickness of the trajectory. The most confident prediction has the least transparent color, the thickest line and the biggest cross.

## D  Additional ablation studies

In Table 3 we ablate different ways to incorporate RPE into the dot-product attention. The differences are insignificant in terms of performance. However, our approach, i.e. adding projected RPE to projected key and value, consumes less memory at training time. We use this setup in our main paper because it can be trained on the RTX 2080 Ti GPUs (12GB VRAM), which are more accessible than the RTX 3090 GPUs (24GB VRAM) in practice.

## E  Additional results

In Tables 4, 5, and 6, we provide the complete results of our HPTR on the WOMD *test* split, the WOMD *valid* split, and the AV2 *valid* split, respectively.

In Figures 9, 10, and 11, we provide more qualitative results on WOMD *valid* of our HPTR predicting vehicles, pedestrians, and cyclists, respectively.

Table 3: Ablation on WOMD valid split. We study different ways to incorporate the RPE into the dot-product attention. Performance is reported as the mean plus-minus 3 standard deviations over 3 training seeds. Models are trained for 60 epochs. OOM: out of memory. $q$: query. $k$: key. $v$: value.

| model description | concat RPE | query RPE | train on 2080ti | mem% on 3090 | Min FDE ↓ | Soft mAP ↑ |
|---|---|---|---|---|---|---|
| ours (add proj. RPE to proj. $k, v$) | × | × | ✓ | 58.2 | $1.143 \pm 0.039$ | $0.401 \pm 0.007$ |
| ours without $q, k, v$ bias | × | × | ✓ | 58.2 | $1.140 \pm 0.021$ | $0.396 \pm 0.009$ |
| add proj. RPE to proj. $q, k, v$ | × | ✓ | OOM | 66.2 | $1.144 \pm 0.036$ | $0.397 \pm 0.006$ |
| concat. RPE to $k, v$ | ✓ | × | OOM | 71.6 | $1.138 \pm 0.026$ | $0.395 \pm 0.006$ |
| concat. RPE to $q, k, v$ | ✓ | ✓ | OOM | 90.5 | $1.133 \pm 0.024$ | $0.396 \pm 0.006$ |

Table 4: Complete results of our HPTR on the WOMD *test* split.

| Object Type | Measurement Time (s) | Soft mAP ↑ | mAP ↑ | Min ADE ↓ | Min FDE ↓ | Miss Rate ↓ | Overlap Rate ↓ |
|---|---|---|---|---|---|---|---|
| Vehicle | 3 | 0.5631 | 0.5475 | 0.2795 | 0.4997 | 0.0927 | 0.0190 |
| Vehicle | 5 | 0.4687 | 0.4623 | 0.5714 | 1.1020 | 0.1297 | 0.0415 |
| Vehicle | 8 | 0.3697 | 0.3664 | 1.0739 | 2.2753 | 0.1787 | 0.0915 |
| Vehicle | Avg | 0.4671 | 0.4587 | 0.6416 | 1.2923 | 0.1337 | 0.0507 |
| Pedestrian | 3 | 0.4534 | 0.4427 | 0.1637 | 0.3111 | 0.0676 | 0.2408 |
| Pedestrian | 5 | 0.3422 | 0.3370 | 0.3220 | 0.6616 | 0.0938 | 0.2648 |
| Pedestrian | 8 | 0.2792 | 0.2751 | 0.5722 | 1.2778 | 0.1248 | 0.2952 |
| Pedestrian | Avg | 0.3582 | 0.3516 | 0.3526 | 0.7502 | 0.0954 | 0.2669 |
| Cyclist | 3 | 0.4334 | 0.4267 | 0.3266 | 0.6078 | 0.1859 | 0.0494 |
| Cyclist | 5 | 0.3587 | 0.3552 | 0.6166 | 1.2085 | 0.1922 | 0.0900 |
| Cyclist | 8 | 0.3025 | 0.3006 | 1.0825 | 2.3096 | 0.2250 | 0.1369 |
| Cyclist | Avg | 0.3649 | 0.3608 | 0.6752 | 1.3753 | 0.2011 | 0.0921 |
| Avg | 3 | 0.4833 | 0.4723 | 0.2566 | 0.4729 | 0.1154 | 0.1030 |
| Avg | 5 | 0.3899 | 0.3848 | 0.5033 | 0.9907 | 0.1386 | 0.1321 |
| Avg | 8 | 0.3171 | 0.3140 | 0.9095 | 1.9543 | 0.1762 | 0.1745 |
| Avg | Avg | 0.3968 | 0.3904 | 0.5565 | 1.1393 | 0.1434 | 0.1366 |

Table 5: Complete results of our HPTR on the WOMD *valid* split.

| Object Type | Measurement Time (s) | Soft mAP ↑ | mAP ↑ | Min ADE ↓ | Min FDE ↓ | Miss Rate ↓ | Overlap Rate ↓ |
|---|---|---|---|---|---|---|---|
| Vehicle | 3 | 0.5611 | 0.5451 | 0.2796 | 0.4988 | 0.0934 | 0.0186 |
| Vehicle | 5 | 0.4704 | 0.4637 | 0.5698 | 1.0986 | 0.1297 | 0.0405 |
| Vehicle | 8 | 0.3678 | 0.3644 | 1.0731 | 2.2909 | 0.1824 | 0.0909 |
| Vehicle | Avg | 0.4664 | 0.4577 | 0.6408 | 1.2961 | 0.1352 | 0.0500 |
| Pedestrian | 3 | 0.4923 | 0.4802 | 0.1454 | 0.2674 | 0.0478 | 0.2358 |
| Pedestrian | 5 | 0.4055 | 0.3993 | 0.2782 | 0.5488 | 0.0661 | 0.2605 |
| Pedestrian | 8 | 0.3639 | 0.3577 | 0.4834 | 1.0157 | 0.0813 | 0.2901 |
| Pedestrian | Avg | 0.4206 | 0.4124 | 0.3023 | 0.6106 | 0.0651 | 0.2621 |
| Cyclist | 3 | 0.4606 | 0.4519 | 0.3309 | 0.6021 | 0.1779 | 0.0523 |
| Cyclist | 5 | 0.3822 | 0.3788 | 0.6136 | 1.1843 | 0.1905 | 0.0897 |
| Cyclist | 8 | 0.2962 | 0.2943 | 1.0661 | 2.3242 | 0.2239 | 0.1433 |
| Cyclist | Avg | 0.3797 | 0.3750 | 0.6702 | 1.3702 | 0.1974 | 0.0951 |
| Avg | 3 | 0.5047 | 0.4924 | 0.2519 | 0.4561 | 0.1064 | 0.1022 |
| Avg | 5 | 0.4194 | 0.4139 | 0.4872 | 0.9439 | 0.1288 | 0.1302 |
| Avg | 8 | 0.3427 | 0.3388 | 0.8742 | 1.8769 | 0.1626 | 0.1748 |
| Avg | Avg | 0.4222 | 0.4150 | 0.5378 | 1.0923 | 0.1326 | 0.1357 |

Table 6: Complete results of our HPTR on the AV2 *test* split.

| minFDE$_6$↓ | minFDE$_1$↓ | minADE$_6$↓ | minADE$_1$↓ | Miss Rate$_6$↓ | Miss Rate$_1$↓ | brier-minFDE$_6$ ↓ |
|---|---|---|---|---|---|---|
| 1.43 | 4.61 | 0.73 | 1.84 | 0.19 | 0.61 | 2.03 |

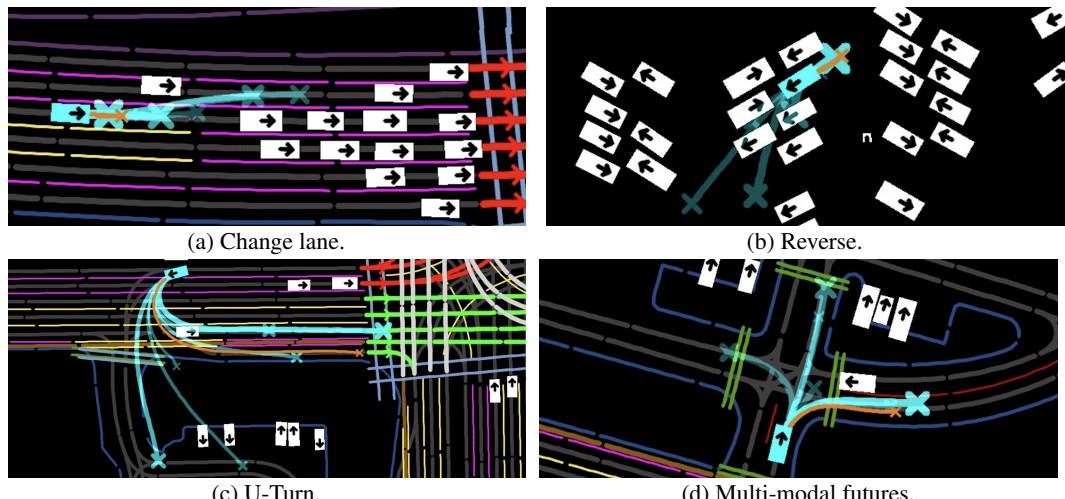

(a) Change lane.

(b) Reverse.

(c) U-Turn.

(d) Multi-modal futures.

Figure 9: Qualitative results of HPTR predicting **vehicles**. Scenarios are selected from the WOMD validation dataset. The ground truth is in orange. The target agent and the predictions are in cyan. The most confident prediction has the least transparent color, the thickest line and the biggest cross.

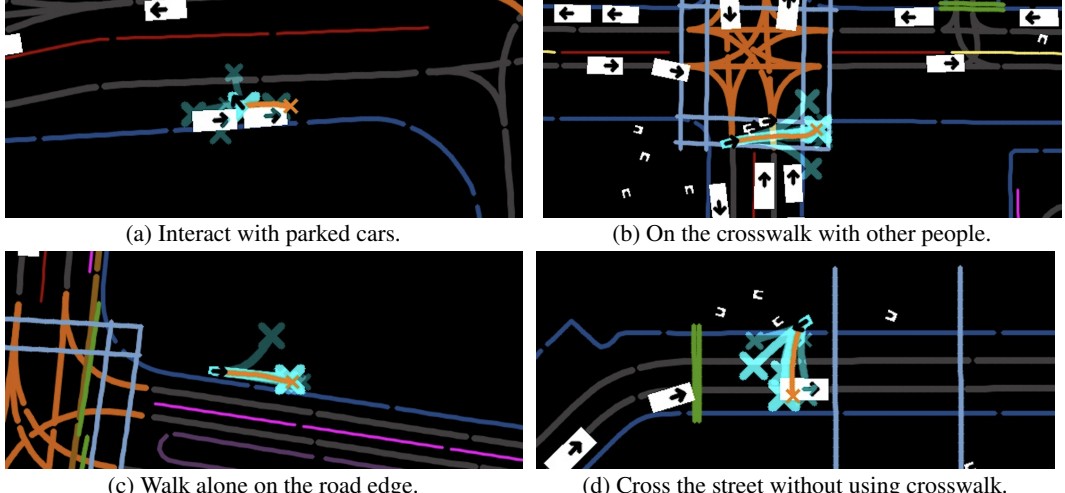

(a) Interact with parked cars.

(b) On the crosswalk with other people.

(c) Walk alone on the road edge.

(d) Cross the street without using crosswalk.

Figure 10: Qualitative results of HPTR predicting **pedestrians**. Read as Figure 9.

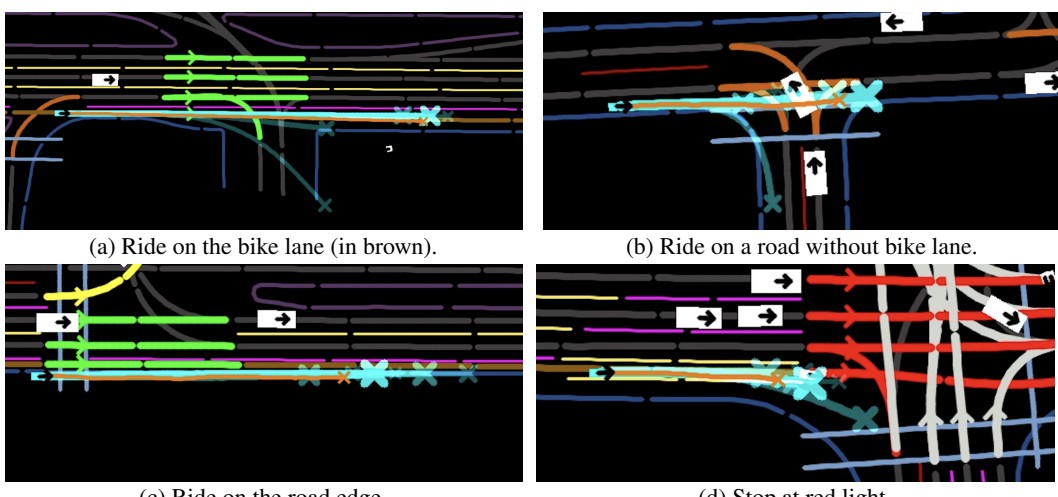

(a) Ride on the bike lane (in brown).

(b) Ride on a road without bike lane.

(c) Ride on the road edge.

(d) Stop at red light.

Figure 11: Qualitative results of HPTR predicting **cyclists**. Read as Figure 9.

