# OpenReview forum: "Real-Time Motion Prediction via Heterogeneous Polyline Transformer with Relative Pose Encoding"
_NeurIPS.cc/2023/Conference — NeurIPS 2023 poster_

### Official Review · Reviewer_LV25 · 2023-06-29

**Soundness:** 3 good
**Presentation:** 3 good
**Contribution:** 2 fair
**Rating:** 6
**Confidence:** 4

**Summary:**

This paper proposes a method to do future motion prediction for autonomous driving agents with the focus of having a computational complexity that is suited for real-time deployment. To do so, they propose a new attention mechanism, called KNARPE, and a hierarchical transformer architecture, called HPTR. By leveraging these components, they achieve SOTA performance (found in agent-centric models) while being close to maintaining the efficiency of scene-centric models. They empirically show this by comparing it to relevant models on both the Waymo and Argoverse 2 datasets.

**Strengths:**

S1) The paper is well-written, has good notation, and is easy to understand. Especially compared to many of the prior works.

S2) Agent-centric approaches may be infeasible for real-time systems because they need one forward-pass per agent. The proposed approach mitigates this issue and reaches the good performance of agent-centric approaches while having a lower computational cost.

S3) The computational cost is rather thoroughly measured, as it looks at GPU memory consumption, offline inference time, and online inference time.

S4) The proposed approach obtains good performance on two standard benchmarks.

S5) There is a theoretical comparison to WayFormer (besides the empirical results).

As a "side-note strength", the authors state an intention to release the code publicly, which should be valuable to the research community.

**Weaknesses:**

W1) The proposed approach is claimed to reach SotA performance at a computational cost that scales in a nice way with the number of agents for which predictions are to be made. There is a thorough comparison to WayFormer, but not to other approaches. For instance, GoRela seems to (very slightly) outperform the proposed approach while also scaling gracefully with additional agents.

W2) The performance of WayFormer in table 2 seems low. Is this some variant of it? In the WayFormer paper, no results are provided on the valid-set, which is what is used for table 2. However, the valid-set seems to be a bit easier than the test-set, so one would expect WayFormer to land a bit above 0.4335 soft mAP, instead of the provided 0.397. Why is this the case?


**Questions:**

Q1) What is Figure 1 contributing to? In my opinion, it is just confusing and lacks proper motivation.

Q2) At l139, should not $t=0$ be included in the history?

Q3) The training takes around 10 days. How is the model convergence, in terms of losses and final KPIs?

Q4) Do other approaches train as long as the proposed approach?

Q5) During experimentation, it is key to revise the method swiftly. If training takes 10 days, did the development of this approach use a shorter schedule for experimentation?

Q6) Based on the development of this approach, do the results of a shorter training schedule seem to correlate well with the results of a longer training schedule?

Q7) How computationally efficient is the proposed approach compared to GoRela? Also see W1.

Q8) See W2.

Q9) The attention to all (all2all) should be a superset of the proposed HPTR architecture. However, as indicated in Table 2, the all2all model is not as good as the HPTR model. Why is this the case? Is there some intuition on this?

Q10) When caching the map features during online inference, these are cached for one timestep, right? I.e., the next timestep the map features are recomputed? Or, are these cached temporally in some way as well?

**Limitations:**

There is a discussion on limitations that provides some additional clarity and insight.

---

> ### Author Rebuttal · Authors · 2023-08-09
>
> Dear Reviewer,
>
> Thank you very much for your helpful comments and suggestions!
> We kindly ask you to read our global response which discusses the comparison with GNN-based pairwise-relative methods and the long training time of our models.
> Now in this post we answer your questions as follows.
>
> ---
> >**Q1**: What is Figure 1 contributing to? In my opinion, it is just confusing and lacks proper motivation.
>
> **A1**:
> We use Figure 1 to demonstrate the limitations of agent-centric approaches and introduce the problem of online inference, both of which are addressed by the approach proposed in our paper.
> At the moment this figure seems to help the understanding of other reviewers.
> We will modify it in the camera-ready if other reviewers have the same concern.
>
> ---
> >**Q2**: At l139, should not t=0 be included in the history?
>
> **A2**:
> Thanks for pointing out! We will fix this in the camera-ready.
>
> ---
> >**Q3**: The training takes around 10 days. How is the model convergence, in terms of losses and final KPIs?
>
> **A3**:
> The training takes 10 days because we have limited GPUs (4 RTX 2080Ti).
> The training time can be reduced given more computational resources.
> In Fig. 2 of the global response PDF, we provided the curves of validation metrics.
> As shown in this figure, the model has largely converged after 5 days of training.
> Therefore, only the models for the final leaderboard submission are trained for 10 days, whereas models for development and ablation are trained for 5 days.
>
> ---
> >**Q4**: Do other approaches train as long as the proposed approach?
>
> **A4**:
> Please refer to the global response for this question.
>
> ---
> >**Q5**: During experimentation, it is key to revise the method swiftly. If training takes 10 days, did the development of this approach use a shorter schedule for experimentation?
>
> **A5**:
> Only the models for final submission are trained for 10 days.
> Other models are trained for 5 days.
> In practice, 5 days are still too long for development.
> But for our cluster, a 4-GPU 5-day job is far easier to be scheduled than an 8-GPU 2-day job.
> Therefore we use this 4-GPU 5-day setup for experimentation.
>
> ---
> >**Q6**: Based on the development of this approach, do the results of a shorter training schedule seem to correlate well with the results of a longer training schedule?
>
> **A6**:
> Fortunately yes.
> In our case training for 5 days is enough for telling the performance of a model as shown in Fig. 2 of the global response PDF.
>
> ---
> >**Q7**:
> How computationally efficient is the proposed approach compared to GoRela? Also see W1.
>
> **A7**:
> Please refer to the global response for this question.
>
> ---
> >**Q8**: See W2. The performance of WayFormer in table 2 seems low. Is this some variant of it? In the WayFormer paper, no results are provided on the valid-set, which is what is used for table 2. However, the valid-set seems to be a bit easier than the test-set, so one would expect WayFormer to land a bit above 0.4335 soft mAP, instead of the provided 0.397. Why is this the case?.
>
> **A8**:
> There are multiple potential reasons.
> Firstly, our model does not apply ensembling, which affects the soft mAP significantly.
> Secondly, Wayformer is not open-sourced, so it could be that our reimplementation is not perfect.
> In fact, our reimplementation has a smaller model size due to the limited computational resources (we use 4 RTX 2080Ti, Waymo uses 16 TPU).
> We will open-source our Wayformer reimplementation such that the community can improve it.
> Thirdly, all models in Table 2 are trained for less epochs because they are meant for ablation studies, not for the final submission.
> And finally, in the Wayformer paper we can still find the performance of their ablation models on the validation split, not in the Tables or in the text but in the Figures (on the y-axis of Fig. 4,5,6 in the Wayformer paper).
> We can see their minADE is never below 0.9, which is way larger compared to their test split submission.
>
> ---
> >**Q9**: The attention to all (all2all) should be a superset of the proposed HPTR architecture. However, as indicated in Table 2, the all2all model is not as good as the HPTR model. Why is this the case? Is there some intuition on this?
>
> **A9**:
> This is because for a fair comparison, the number of layers of models in Table 2 are selected such that their total numbers of learnable parameters are roughly the same (15M in our case).
> As a consequence, the all2all model in Table 2 is not as deep as other models.
> Given more layers and longer training time, the all2all model can reach the same performance as our HPTR.
> It is intuitive that the all2all model is more difficult to train because without any inductive bias it allows all possible attentions.
> Our point here is to show the hierarchical architecture we proposed can achieve the same performance with less parameters and training time.
>
> ---
> >**Q10**: When caching the map features during online inference, these are cached for one timestep, right? I.e., the next timestep the map features are recomputed? Or, are these cached temporally in some way as well?
>
> **A10**:
> During the online inference, the encoded map features are computed only once at the beginning $t=0$ and cached.
> At $t=0$ we will predict the future trajectories at $t={1,2,\dots, T}$.
> At the next time step $t=1$ we will reuse the cached map features and predict the future trajectories at $t={2,3,\dots, T+1}$.
> The cached map features can be reused because we assume the map is static, i.e. it does not change from $t=0$ to $t=1$.
> In our online inference experiments (right plot Fig. 4), the map features cached at $t=0$ are reused for 100 times (i.e. 10 seconds) and we compute the average latency over these 100 inferences.
> In this case we assume the map does not change within 10 seconds.
> In the real world we can reuse the cached map features for a longer period of time, say days or maybe weeks.
> Recomputing the map features is necessary when the HD maps are changed, which does not happen very frequently.

---

> > ### Comment · Reviewer_LV25 · 2023-08-14
> >
> > Thank you for the thorough answers. I have read the other reviews as well together with their rebuttals. In my opinion, this paper would be a valuable addition to the machine learning community.

---

### Official Review · Reviewer_753q · 2023-06-29

**Soundness:** 3 good
**Presentation:** 3 good
**Contribution:** 3 good
**Rating:** 6
**Confidence:** 4

**Summary:**

This paper proposes a motion prediction framework HPTR. As agent-centric presentation usually has a high computational cost and poor scalability, the paper uses the transformer to encode pairwise-relative representation with K-nearest neighbor attention and relative pose encoding. It proposes a hierarchical transformer-based framework to efficiently encodes intra-class and inter-class information which allows asynchronous update for better online inference efficiency. Experiments on Waymo Open Motion Dataset and Argoverse-2 motion dataset show its superior performance and good efficiency.

**Strengths:**

1. The proposed hierarchical transformer-based framework is efficient and enables asynchronous token update which is usually ignored in other motion prediction works.
2. The paper adopts relative pose encoding to better unleash the expressiveness of the pairwise-relative representation.
3. Experiment results have shown that the proposed method has achieved a good balance between performance and efficiency.
4. The overall writing is clear and easy to follow.

**Weaknesses:**

1. It is claimed in the paper that this paper uses transformer and pairwise-relative representation which is less computational than GNN. However, GNN and transformer can be viewed as equivalent if attention is used to aggregate and update information among the nodes as in HDGT. A more in-depth analysis should be provided to better clarify the difference and advantages of the proposed method over GNN based method with pairwise-relative representation like HDGT.
2. ProphNet also achieves very good results in terms of both performance and efficiency. And in ProphNet's paper, the single model (without ensembling) has achieved 1.89 brier-min FDE6 on the AV2 dataset. The author should include this result and compare the proposed with it in the efficiency part as well. And the analysis of the comparison is also important.
3. As previous pairwise-relative representation methods do not use pairwise-relative representation, the ablation experiment on RPE should be included to analyze how much performance gain is due to RPE.

**Questions:**

As the proposed method is built upon transformer blocks and the transformer is well-recognized for its good scalability, have the authors tried to scale up the model to see its performance improvements?

**Limitations:**

The authors have well discussed the limitation in Sec5.

---

> ### Author Rebuttal · Authors · 2023-08-09
>
> Dear Reviewer,
>
> Thank you very much for your helpful comments and suggestions!
> We kindly ask you to read our global response which discusses the comparison with GNN-based pairwise-relative methods and the long training time of our models.
> Now in this post we answer your questions as follows.
>
> ---
> >**Q1**: It is claimed in the paper that this paper uses transformer and pairwise-relative representation which is less computational than GNN. However...
>
> **A1**:
> We agree that Transformer can be formulated as a special case of GNN and theoretically attention mechanism is not computationally more efficient than message passing.
> However, in practice, most of the time Transformers are more efficiently implemented on GPUs compared to GNNs.
> As shown in Fig. 1 of the global response PDF, KNARPE is implemented with the most basic matrix operations (matrix indexing, summation and element-wise multiplication).
> Based on KNARPE, our HPTR uses only these basic matrix operations which are easier to be efficiently deployed than most of the GNNs.
> In terms of performance, our method outperforms HDGT by a large margin (cf. Table 2 and the WOMD leaderboard).
> In terms of efficiency, our method is faster than HDGT by an order of magnitude (cf. Fig. 3 of the global response PDF).
> Please refer to the global response for the detailed discussion.
>
> ---
> >**Q2**: ProphNet also achieves very good results in terms of both performance and efficiency. And in ProphNet paper, the single model...
>
> **A2**:
> We agree that ProphNet (CVPR 2023) as a concurrent work achieves excellent performance and efficiency on AV1 and AV2 datasets.
> According to Table 3 of the ProphNet paper, it achieves 1.89 brier-min FDE6 on the AV2 dataset with a single model.
> According to Sec. 4.3 of the ProphNet paper, that single model "uses 3 heads and each with 6 output trajectories".
> According to Sec. 3.6 of the ProphNet paper, ProphNet "generates more proposals than the required number of output modality".
> Therefore, we believe it is fair to claim ProphNet predicts more futures than required, whereas it is controversial to claim ProphNet uses ensembling because the single model ensembles only part of its network, i.e. the hydra heads.
> The performance of ProphNet in Table 1 of our paper was obtained from the AV2 leaderboard accessed early this year when we wrote our paper.
> That submission entry is now removed from the AV2 leaderboard, but the performance can still be found in the appendix of the ProphNet paper (Table 7).
> According to the appendix A of the ProphNet paper, they "train three different models for ensembling".
> So the ProphNet in the Table 1 of our submission actually used ensembling.
> In the camera-ready we will update Table 1 and replace the old ProphNet with ensembling by the new single-model ProphNet.
> We will also remove the dagger in front of ProphNet as the single-model ProphNet is not exactly an ensembling.
>
> Since ProphNet is a concurrent work and it is not open-sourced, we don't have enough time to reproduce it and provide a detailed efficiency comparison in our submission.
> However, since ProphNet is still agent-centric, the scalability problem is inevitable.
> We hope the comparison with Wayformer (ICRA 2023), another SOTA agent-centric method, should be enough to demonstrate the advantage of our method over general agent-centric methods.
> Moreover, according to Table 4 and Sec. 4.5 of the ProphNet paper, they have 28ms latency per agent (64 agents in total), which means per episode the latency is 1792ms.
> This is more than a magnitude slower than our approach which has 60ms latency during the offline inference with 64 agents.
>
> ---
> >**Q3**: As previous pairwise-relative representation methods do not use pairwise-relative representation, the ablation experiment on RPE should be included to analyze how much performance gain is due to RPE.
>
> **A3**:
> We think there is a typo in this question, it should be "As previous pairwise-relative representation methods do not use ~pairwise-relative representation~ RPE, the ablation experiment on RPE should be included to analyze how much performance gain is due to RPE.".
>
> Our answer is as follows.
> RPE can be ablated from different aspects.
> Firstly, we can ablate the "relative" aspect, i.e. instead of the pairwise-relative representation we use the agent-centric or scene-centric representation.
> This ablation has been done in Table 2 of our paper.
> Now given the pairwise-relative representation, there are different ways to use it.
> Our specific methods RPE and KNARPE are defined respectively in Eq. 1-3 and Eq. 4-5 of our paper.
> Eq. 1-3 are the standard positional encoding which is applied to almost all Transformer-based methods.
> Since it is a common practice to use positional encoding to pre-process the inputs to Transformer blocks, we omit its ablation in our paper.
> In Table 1 of our appendix we have provided detailed ablations on Eq. 4-5, i.e. different attention mechanisms for RPE.
> We show that our KNARPE achieves the best performance and efficiency.
> Back to the question "previous pairwise-relative representation methods do not use RPE", this is because all prior works are based on GNNs and our RPE is specifically designed for Transformers.
> Combining GNNs with RPE (Eq. 1-3) is an interesting idea but it is out of the scope of our paper which focuses on Transformers.
>
> ---
> >**Q4**: As the proposed method is built upon transformer blocks and the transformer is well-recognized for its good scalability, have the authors tried to scale up the model to see its performance improvements?
>
> **A4**:
> We do observe a proportional relationship between the performance and the model size.
> However, due to the limited computational resources (4 RTX 2080Ti), we cannot experiment with a larger model size without further reducing the batch size, which is already rather small (B=12) at the moment.

---

> > ### Comment · Reviewer_753q · 2023-08-19
> > **Response to the rebuttal**
> >
> > Thank you for the detailed responses. My main concerns have been resolved and I would like to raise my rating to weak accept.

---

### Official Review · Reviewer_Ccz6 · 2023-07-04

**Soundness:** 2 fair
**Presentation:** 3 good
**Contribution:** 2 fair
**Rating:** 4
**Confidence:** 4

**Summary:**

This work proposes a novel method for motion forecasting which uses an efficient attention mechanism with pairwise relative representation and asynchronous updates for the static & dynamic parts of the scene. Extensive experiments on Waymo and Argoverse datasets show competitive performance to existing methods while being more efficient than agent-centric methods.

**Strengths:**

- This work incorporates a pairwise relative representation in a k-nearest neighborhood for attention mechanism which is more efficient than quadratic attention.
- The proposed method minimizes redundancies in computation by sharing context among agents and using asynchronous updates for static and dynamic tokens. This makes it as efficient as scene-centric methods.
- The proposed method is competitive to existing approaches (Table 1) on Waymo and Argoverse datasets which do not use ensembles, while more efficient in terms of memory consumption and inference latency.
- The ablations (Table 2) are helpful in understanding the benefits of different components in the proposed method. Also, mean & standard deviation over 3 runs are reported to account for the randomness in training.

**Weaknesses:**

- For pairwise relative poses, it'd be useful to compare with a simpler alternative of using relative distances directly (instead of sinusoidal encoding), as done in Interaction Transformer (eq 4. in [1]).
- How does pairwise-relative representation retain the good scalability of scene-centric representation? Doesn't the scalability come from k-nearest neighbors which reduces the number of agents considered for attention?
- L165 states that the performance deteriorates if the input is scene-centric. Why is this the case? The pairwise relative representation should help with standard self-attention as well.
- What is the difference between the middle and right plots in Fig. 4?
- It'd be useful to have efficiency comparison against other approaches, eg. GoRela since it also uses pairwise relative representation and achieves similar performance.
- MTR-e2e and GoRela have similar performance to the proposed approach in Table 1. Without efficiency comparisons with these methods, the performance benefits are not clear.
- Does the WF baseline in Table 2 use ensembles?

[1] Li et al. End-to-end Contextual Perception and Prediction with Interaction Transformer. IROS 2020

**Questions:**

The main concern is that the performance benefits of the proposed approach are not clear. MTR-e2e and GoRela have similar performance to the proposed approach and efficiency comparisons with these methods are not provided. Other clarifications required are mentioned in the weaknesses above.

**Limitations:**

Limitations are discussed.

---
I have read the rebuttal, other reviews, and discussion. I appreciate the additional ablations and clarifications provided by the authors. While I agree that efficiency is an important consideration, I am not convinced about the effectiveness of the proposed approach. Since efficiency is the central claim, I would expect to see clear gains over baselines that use some combination of vectorized inputs, pairwise relative information, and/or transformer in their architecture. Looking at the results in Wayformer & HiVT papers, they report latency in the range of 30-60ms for different variants which is in the similar range as HPTR. I think the results should be shown in the form of performance vs latency vs capacity plots (similar to Fig. 4,5,6 in Wayformer paper) while comparing to different baselines to show the benefits of HPTR. So, I am retaining my rating of `Borderline reject`.

---

> ### Author Rebuttal · Authors · 2023-08-09
>
> Dear Reviewer,
>
> Thank you very much for your helpful comments and suggestions!
> We kindly ask you to read our global response which discusses the comparison with GNN-based pairwise-relative methods and the long training time of our models.
> Now in this post we answer your questions as follows.
>
> ---
> > **Q1**: For pairwise relative poses, it'd be useful to compare with a simpler alternative...
>
> **A1**:
> Unfortunately we cannot provide additional experimental results during the rebuttal phase because the cluster of our institution has been undergoing maintenance.
> However, we can discuss this idea from a theoretical perspective.
> In contrast to the relative pose, the relative distance does not contain the necessary information for making driving decisions.
> It only tells us how far away an object is, but not at which direction it is located with respect to the agent of interest.
> Therefore, replacing relative poses with relative distances would not work in our case.
> Given distances and orientations, it is common practice to use sinusoidal positional encoding (such as Eq. 1-3 in our paper) to pre-process the inputs to Transformer blocks.
> Nowadays, almost all SOTA Transformers have the positional encoding in their design.
> Nevertheless, we think this is an interesting idea and we will add this discussion to the camera-ready.
>
> ---
> > **Q2**: How does pairwise-relative representation retain the good scalability...
>
> **A2**:
> Let us take a lane segment as an example.
> Say we have N agents to be predicted.
> Agent-centric methods will transform this single lane segment to the coordinate of each of the N agents.
> Effectively, we will have N copies of the same lane segment and we will encode the same segment N times.
> Sene-centric methods present the lane segment in the global coordinate, hence it is encoded only once.
> Pairwise-relative methods decompose the lane segment into a high-dimensional local attribute ($dim\gg3$), and a low-dimensional global pose ($dim=3$).
> The high-dimensional local attribute is encoded only once and shared by all agents, hence the good scalability.
> The 3D global poses will be used to compute the 3D relative poses between the lane segment and N agents.
> To conclude, both the KNN and the pairwise-relative representation contributes to the scalability.
> Considering $N$ tokens and hidden dimension $D$, KNN reduces the complexity from $\mathcal{O}(N^2D)$ to $\mathcal{O}(NKD)$ by restricting the attention of each token to its K-nearest neighbors.
> The pairwise-relative representation reduces the complexity from $\mathcal{O}(N^2D)$ to $\mathcal{O}(N^2\cdot 3+ND)=\mathcal{O}(N^2\cdot 3)$ by sharing the high-dimensional local attribute.
>
> ---
> > **Q3**: L165 states that the performance deteriorates if...
>
> **A3**:
> This question includes two parts.
>
> Firstly, why the performance deteriorates if the input is scene-centric?
> The representation $(p_i,u_i)$ in L165 is scene-centric because the global pose $p_i$ is in the global coordinate.
> Using $(p_i,u_i)$ directly as input to the standard self-attention is investigated in SceneTransformer, which is outperformed by other methods by a large margin because scene-centric representation is not rotation and translation invariance.
>
> Secondly, shouldn't the pairwise-relative representation help with standard self-attention as well?
> In Eq. 1-5 we propose a new attention mechanism to process $(r_{ij}, u_i)$ .
> In Sec. C of our appendix we have ablated other variations of attention mechanism for the pairwise-relative representation.
> Since the standard self-attention cannot process the pairwise relative representation, we experimented with some variations which are very similar to the standard self-attention.
> The results show that using pairwise relative representation with (a moderately modified version of) standard self-attention outperforms scene-centric methods, but it is not as good as the KNARPE we proposed.
>
> ---
> > **Q4**: What is the difference between the middle and right plots in Fig. 4?
>
> **A4**:
> The middle plot is offline inference and the right plot is online inference.
> Offline means doing inference with datasets, whereas online means doing inference with streaming inputs, as if on a real car.
> During offline inference, for each episode we will inference only once at $t=0$.
> During online inference, for each episode we will inference consecutively at $t=\{0,1,\dots,T\}$, where $T=99$ in Fig. 4.
> Our HPTR allows the encoded static map features to be reused across these time steps during the online inference, hence the latency of our models in the right plot is lower than in the middle plot.
> We will try to improve the clarity of the caption of Fig. 4 in the camera-ready.
>
> ---
> > **Q5**: It'd be useful to have efficiency comparison against other approaches, eg. GoRela...
>
> **A5**: Please refer to the global response.
>
> ---
> > **Q6**: MTR-e2e and GoRela have similar performance to the proposed approach in Table 1...
>
> **A6**:
> As shown in Table 1 (WOMD valid), our method outperforms MTR-e2e substantially in mAP (0.415 vs. 0.3245), which is the major metric considered by the WOMD leaderboard.
> On the AV2 dataset, our method is on a par with GoRela.
> However, GoRela focuses on AV2 and does not provide any results on WOMD, whereas we focus on WOMD and tune all hyperparameters for WOMD.
> We believe our performance on the AV2 leaderboard could be further improved given sufficient tuning on the AV2 dataset.
> In terms of efficiency, Wayformer is one of the most efficient agent-centric methods.
> Since MTR-e2e is also agent-centric and it is slower than Wayformer (cf. the appendix of MTR and the results section of Wayformer) which we clearly outperform in terms of efficiency, we believe an efficiency comparison with MTR-e2e is unnecessary.
> For the efficiency comparison with GNN-based pairwise-relative methods such as GoRela, please refer to the global response.
>
> ---
> > **Q7**: Does the WF baseline in Table 2 use ensembles?
>
> **A7**: No, none of the models in Table 2 uses ensembles.

---

> > ### Comment · Reviewer_Ccz6 · 2023-08-14
> > **Response to rebuttal**
> >
> > I have read the rebuttal and other reviews. I appreciate the additional comparisons and clarifications provided by the authors, which helped me get a better understanding of the paper. I also went through the related work again to identify the differences between the proposed work and existing methods.
> >
> > From my understanding, the central claim of the paper is efficiency. To achieve this, 2 components are proposed - KNARPE and HPTR. I need some more clarification regarding these:
> >
> > **KNARPE**: It has 2 main parts - K-nearest neighbor attention and pairwise relative pose encoding.
> > - K-nearest neighbor attention: This is computed based on L2 distance (L169) and the value of K is different for different transformers (L248-252), going upto K=360 for AC-to-all transformer. Since L2 distance is used, this can also be considered as applying attention in a neighborhood of a certain radius. This has been done in previous works, eg. HiVT [70]. In this regard, can K-nearest neighbor attention be interpreted as attention over a local region (as in HiVT)? Or are there any significant differences between the two?
> > - Pairwise relative pose encoding: This takes into account the relative translation and orientation between different entities in the scene. This has been considered in prior work, eg. HiVT in agent-agent, agent-lane, and global interaction modules, which also uses attention & transformers in the architecture. Are there any significant differences between the pairwise relative encodings in the proposed work and HiVT?
> >
> > **HPTR**: It involves a polyline transformer architecture and asynchronous updates of heterogeneous tokens (map features are cached).
> > - The architecture consists of a transformer applied to vectorized inputs (in the form of polylines) with relative pairwise encodings. This is similar to HiVT architecture.
> > - The main difference is the asynchronous token updates. While the map features are cached, the main bottleneck in computation would come from AC-to-all transformer since it contains the most tokens (K=360). Is this correct?
> >
> > **Results**
> > Since the central claim is efficiency, the most important experiment is efficiency analysis. In Sec 4.4, it is stated that HPTR can make predictions in 37 ms, which can be reduced to 25 ms (40 fps) with better implementation. From the inference speed results in HiVT (Sec 4.3 and Table 5), it seems like HiVT can also run real-time with similar latency as the proposed approach. Since the experiment settings are different in the 2 papers, it might be hard to compare the two inference speed directly. It'd be helpful if the authors can provide more insights into the runtime & performance comparison between HPTR and HiVT (since it seems to be the most relevant baseline).

---

> > > ### Author Response · Authors · 2023-08-16
> > > **Answers to the questions regarding HiVT**
> > >
> > > Dear Reviewer Ccz6,
> > >
> > > Thanks for your comments.
> > > We are glad that our rebuttal has addressed your previous concerns.
> > > In the following we answer your questions regarding HiVT.
> > > >Q1: Can KNN attention be interpreted as attention over a local region (as in HiVT)?
> > >
> > > Yes, the KNN attention can be interpreted like that.
> > > However, HiVT uses a distance threshold for selecting neighbors, whereas we use a threshold directly on the number of neighbors.
> > > This leads to significant differences in the implementation in practice, because the distance threshold does not enforce an upper bound on the number of neighbors.
> > > As a consequence, the Transformers of HiVT are implemented with message passing and GNN libraries, which are less efficient than our HPTR implemented with basic matrix operations.
> > > This highlights the importance of the KNN design in our KNARPE module.
> > > >Q2: Are there any significant differences between the pairwise relative encodings in the proposed work and HiVT?
> > >
> > > As mentioned in L93 of our paper, the most fundamental difference between HiVT and our method is that HiVT considers vectors whereas our HPTR considers polylines.
> > > The pairwise-relative polyline representation boils down to an agent-centric representation if polylines are singletons, i.e. vectors.
> > > As a result, the local encoders of HiVT are actually agent-centric.
> > > It is a bit confusing because HiVT formulates its inputs in a pairwise-relative way, but essentially it is agent-centric.
> > > To verify this, we can observe HiVT does not share information among agents or across time steps, similarly to other agent-centric methods.
> > > Given a new vector, the local encoders of HiVT transform the vector to the local coordinate of each agent.
> > > Nevertheless, HiVT is still closely related to our method because its global interaction module follows the concept of pairwise-relative representation.
> > > HiVT can be seen as an agent-centric method augmented with a pairwise-relative module (the global interaction) during the decoding phase in order to realize multi-agent prediction.
> > > As shown in Table 1 of the HiVT paper, without the global interaction module, the pure agent-centric HiVT could still achieve reasonable performance.
> > > >Q3: The architecture consists of a transformer applied to vectorized inputs with relative pairwise encodings. This is similar to HiVT architecture.
> > >
> > > HiVT and our HPTR differ fundamentally in terms of how to apply Transformer to vectorized inputs with relative poses.
> > > As stated in L93 in our paper, HiVT uses the standard Transformer, whereas we proposed our own attention mechanism.
> > > One of our main contributions is the attention mechanism defined in Eq. 1-5 of our paper.
> > > In contrast to our method, HiVT does not compute the RPE as we did in Eq. 1-3; it rather concatenates the relative poses directly with other attributes.
> > > After that, HiVT uses the concatenated tensors as the input to the standard attention; it does not propose a new attention mechanism as we have done in Eq. 4-5.
> > > In our appendix we have ablated HPTR using the standard attention, i.e. we do not apply Eq. 4-5 similar to HiVT.
> > > As shown in Table 1 of our appendix, this does not improve the performance but significantly increase the demand of computational resources.
> > > >Q4: While the map features are cached, the main bottleneck in computation would come from AC-to-all transformer...
> > >
> > > Yes, the main bottleneck comes from the Transformer block that contains the most layers and the largest attention matrix.
> > > >Q5: Why HiVT is not considered in the experiments?
> > >
> > > For two reasons.
> > > Firstly, HiVT reports performance only on AV1, which is outdated and has been replaced by AV2.
> > > Secondly, HiVT has been outperformed by many publications by a large margin on AV1.
> > > The rankings on the AV1 leaderboard are 35th for HiVT, 17th for MultiPath++, and 6th for Wayformer.
> > > According to Table 1 of our paper, our HPTR is on a par with MultiPath++ on WOMD.
> > > Since we have compared with the more recent SOTA methods on the most recent and challenging datasets, we think it is redundant to consider HiVT in our experiments.
> > >
> > > In terms of run-time, HiVT does not focus on efficiency and scalability.
> > > The latency of HiVT is close to that of our method, but HiVT considers a simpler dataset (AV1 vs. WOMD) and it uses fewer parameters (2.5M vs. 15M) and a small perceptive field (50m) in its implementation, all of which reduce latency but not the algorithmic complexity.
> > > Therefore, as the reviewer has pointed out, it is hard to compare the inference speed directly.
> > > However, since HiVT is essentially agent-centric, it still suffers fundamentally from poor scalability due to its higher complexity.
> > >
> > > ---
> > > To conclude, our contributions are still valid and novel, as they are not presented in the HiVT paper.
> > > Compared to GoRela and HDGT which we have thoroughly examined in our initial rebuttal, HiVT is less relevant of a baseline for our method.
> > > Nevertheless, we think this discussion about HiVT is very intriguing and we will add it to our appendix.

---

### Official Review · Reviewer_Gjvi · 2023-07-06

**Soundness:** 3 good
**Presentation:** 3 good
**Contribution:** 3 good
**Rating:** 6
**Confidence:** 4

**Summary:**

This paper introduces several ideas to boost the efficiency of marginal motion prediction: (1) represent all input entities as polylines without global pose attributes, (2) use transformer architectures but limit attention to K nearest neighbors, (3) directly use relative pose in transformer computations, (4) apply full self-attention only to map tokens, which can be cached during online inference, (5) obtain traffic light and agent features hierarchically, with cross-attention, and (6) use a final cross-attention block for all agent-anchor pairs to directly decode trajectories without any clustering or ensembling. All of these ideas are intuitive and an ablation study discusses some of their individual contributions. The final model obtains reasonable performance on WOMD and Argoverse 2 while scaling to dense traffic much more feasibly than one of the existing SoTA methods, Wayformer.

**Strengths:**

The key contribution of this work lies in clearly highlighting of some problematic practices which are still commonly used in most research on motion forecasting in autonomous driving (heavy emphasis on the offline setting), and bringing efficiency for online inference to the forefront. The ideas presented to improve efficiency are not all new, but in combination interesting and well-motivated. Despite the large number of complex technical concepts covered in the draft, the presentation is clear and it is possible to follow and understand all components.

**Weaknesses:**

While all the proposed ideas are simple and intuitive, putting them all together yields a complex architecture with a large space of design choices and hyper-parameters. This model trains for 10 days, despite the efficient vectorized input and hierarchical architecture focused on efficiency. The runtime analysis only presents a comparison to an agent-centric baseline Wayformer, which fails to provide evidence for whether the proposed model is efficient among relative pose based forecasting methods (e.g., no evidence for the claim made in L092 that GNNs are more demanding than transformers in the online inference setting).

**Questions:**

1. Is the training time of HTPR similar to Wayformer for the same number of training epochs?
2. How is the KNARPE operation implemented in practice? Do you still compute and mask a dense attention matrix, or implement custom kernels to only compute attention where needed?
3. How important is the post-processing described in L254-257? Are these techniques commonly applied by methods on these leaderboards?
4. Could you please elaborate on L261-262, what does sampling 25% and 50% mean in this context?
5. Would it be possible to compare the inference time (Fig. 4) to a GNN method with relative pose encodings?

Minor:

1. Have you tried adding the blue dots from Fig. 1b to Fig. 1a as well? This could make it clearer to understand which agents are being used in Fig. 1b.
2. Could Fig. 5 be simplified, in particular by removing the striking colors for the map elements? An alternative option would be to add a legend describing all colors.

Update:

Thank you for the detailed responses to all questions. The rebuttal addresses all of my concerns, and I would like to maintain my positive rating.

**Limitations:**

Limitations are discussed in Section 5.

---

> ### Author Rebuttal · Authors · 2023-08-09
>
> Dear Reviewer,
>
> Thank you very much for your helpful comments and suggestions!
> We kindly ask you to read our global response which discusses the comparison with GNN-based pairwise-relative methods and the long training time of our models.
> Now in this post we answer your questions as follows.
>
> ---
>
> > **Q1**: Is the training time of HTPR similar to Wayformer for the same number of training epochs?
>
> **A1**:
> No.
> Each training epoch of HPTR takes longer time than Wayformer does.
> Given 5 days of training time, we can train our reimplementation of Wayformer for 110 epochs, whereas for HPTR it is 60 epochs.
> However, HPTR is more sample efficient.
> As shown in Table 2 of our submission, the performance of Wayformer at epoch 100 is roughly the same as HPTR at epoch 60.
> So as a result, HPTR and Wayformer converge roughly at the same speed if measured in wall time.
> The training of HPTR could be further accelerated by pre-computing and saving the relative poses during the dataset pre-processing.
>
> ---
>
> > **Q2**: How is the KNARPE operation implemented in practice? Do you still compute and mask a dense attention matrix, or implement custom kernels to only compute attention where needed?
>
> **A2**:
> We implement our custom multi-head attention with matrix indexing, summation and element-wise multiplication.
> Given $src\in \mathbb{R}^{B\times M\times D}$ and $tgt\in \mathbb{R}^{B\times N\times D}$, where $B$ is the batch size, $M$ is the length of $src$, $N$ is the length of $tgt$ and $D$ is the hidden dimension.
> The first step is to get the K-nearest-neighbor $tgt$ for each $src$, i.e. get $\text{tgt}_{knn} \in \mathbb{R}^{B\times M \times K \times D}$ by indexing $tgt$ based on the L2 distances $dist \in \mathbb{R}^{B\times M\times N}$ between $src$ and $tgt$.
> After that, we use element-wise multiplication instead of matrix multiplication to compute the attention matrix $A \in \mathbb{R}^{B\times M\times K}$.
> All tensors have fixed shape in our implementation and we use masking to address missing tokens.
> Figure 1 in the PDF of the global response illustrates in detail how KNARPE is implemented in practice.
> We will add this figure and the implementation details of KNARPE to our appendix.
>
> ---
>
> > **Q3**: How important is the post-processing described in L254-257? Are these techniques commonly applied by methods on these leaderboards?
>
> **A3**:
> The non-maximum suppression post-processing can improve the mAP and soft mAP significantly.
> It is used by almost all methods submitted to the WOMD leaderboard.
> Our specific implementation follows MPA [1], which is one of the winners of the WOMD challenge 2022.
>
> ---
>
> > **Q4**: Could you please elaborate on L261-262, what does sampling 25\% and 50\% mean in this context?
>
> **A4**:
> We apologize for the confusion.
> This sampling is just because we want to perform validation, metrics logging and checkpoint saving more frequently.
> By sampling 50\% of the training dataset at each epoch, we effectively reduce the duration of each training epoch by 50\% but still ensure that all data from the training split is used for training.
> This is not a necessary step.
> Training for 60 epochs while sampling 50\% of the training dataset at each epoch is equivalent to training for 30 epochs while using the complete training split at each epoch.
> The difference is negligible in our case because our training runs for many epochs and all samples from the dataset are alike (they are from the same domain).
> The only effective difference is that the former logs the validation metrics twice as often as the latter, and the parameters of the learning rate scheduler should be changed accordingly if it schedules based on epoch numbers.
>
> ---
>
> > **Q5**: Would it be possible to compare the inference time (Fig. 4) to a GNN method with relative pose encodings?
>
> **A5**: Please refer to the global response for this question.
>
> ---
>
> **Others**:
> Thanks for the suggestions.
> We will add the blue dots from Fig. 1b to Fig. 1a in the camera-ready.
> We will also try to make Fig. 5 more readable in the camera-ready.
> Due to the limited space in the main paper, we will add the explanation of the visualization of Fig. 5 to the appendix.
>
> ---
> [1] Stepan Konev. Mpa: Multipath++ based architecture for motion prediction. arXiv preprint arXiv:2206.10041, 2022

---

> > ### Comment · Reviewer_Gjvi · 2023-08-19
> >
> > Thank you for the detailed responses to all questions. The rebuttal addresses all of my concerns, and I would like to maintain my positive rating.

---

### Official Review · Reviewer_yHiV · 2023-07-13

**Soundness:** 3 good
**Presentation:** 2 fair
**Contribution:** 2 fair
**Rating:** 3
**Confidence:** 4

**Summary:**

--

**Strengths:**

--

**Weaknesses:**

--

**Questions:**

--

**Limitations:**

--

---

> ### Author Rebuttal · Authors · 2023-08-09
>
> Since this review does not provide any detailed comments, we will omit the rebuttal in this case.

---

### Author Rebuttal · Authors · 2023-08-09

## Global Response

We thank all reviewers for their helpful feedback.
We are glad that the reviewers appreciate our technical contributions, specifically the KNARPE attention mechanism, the asynchronous token update and the efficiency comparison.
Moreover, we are happy to see that the reviewers agree with us on the importance of identifying and solving the efficiency problems of motion prediction in autonomous driving.

In this global response, we will address two concerns raised by multiple reviewers.

---

### 1. The efficiency comparison with other methods based on GNN and pairwise-relative representation.

To the best of our knowledge, currently there are only two such methods, GoRela and HDGT.
GoRela is published on ICRA 2023 and it is not open-sourced, so we would need to reimplement it.
However, the hyperparameters of its network, such as the hidden dimension, number of layers of each component etc, are not disclosed in the GoRela paper.
Without this information, we cannot reproduce its performance and conduct a fair efficiency comparison.
After receiving the NeurIPS reviews, we immediately sent an Email to the authors of GoRela asking for these implementation details, but we haven't received any response yet.
As such, we cannot provide an efficiency comparison with GoRela during this rebuttal phase.
In terms of performance, our method is on par with GoRela on the AV2 dataset.
However, GoRela focuses on AV2 and it does not provide any results on WOMD, whereas we focus on WOMD and tune all our hyperparameters on WOMD.
We believe our performance on the AV2 leaderboard could be further improved given sufficient tuning on the AV2 dataset.

Fortunately, two weeks ago on 2023-07-20 HDGT was open-sourced.
HDGT is published in CoRL 2022 and TPAMI 2023.
It does not perform as good as our method and GoRela, but since it is based on GNN and pairwise-relative representation, we believe the efficiency comparison with HDGT should address the reviewers' concern effectively.
As shown in the left plot of Fig. 3 in the global response PDF, HDGT demonstrates good scalability in terms of GPU memory consumption because it uses the pairwise-relative representation.
As shown in the middle plot of Fig. 3, in terms of offline inference speed, HDGT is slower than our HPTR, and it is actually even slower than our agent-centric Wayformer baseline.
To confirm that HDGT runs correctly on our setup, in the right plot of Fig. 3 we reproduce the inference time of HDGT on the complete WOMD validation split with different validation batch size and we compare the reproduce numbers with the reported number in the TPAMI paper.
The slow inference speed of GNN-based methods such as HDGT is mainly because the GNN libraries cannot utilize the GPU as efficiently as the basic matrix operations do.
As shown in Fig. 1 of the global response PDF, our KNARPE is implemented with the most basic matrix operations (matrix indexing, summation and element-wise multiplication), hence it is better suited for real-time and on-board applications.

---

### 2. The long training time of our models.

Our final models for the leaderboard submission are trained for 10 days and models for ablation and development are trained for 5 days.
This long training time is because on the one hand WOMD is a very large-scale dataset, and on the other hand we only use 4 RTX 2080Ti GPUs for the training.
While comparing the wall time duration of training, the computational resources should be taken into consideration.
As a reference, HDGT uses 8 V100 and trains for 4-5 days, GoRela uses 16 GPUs (model not specified but most likely A100/V100), MTR uses 8 RTX 8000, Wayformer uses 16 TPU v3 cores and ProphNet uses 16 V100.
All of these methods use a much higher number of more powerful GPUs than we use.
If we had their computational resources, the training time of our method could be reduced to 1-2 days.
In terms of sample efficiency, our method is on par with other methods.
As shown in Fig. 2 of the global response PDF, our HPTR converges after 15 epochs (5 days) and our final model is trained for 30 epochs (10 days) on WOMD.
As a reference, HDGT is trained for 30 epochs, MTR is trained for 30 epochs and ProphNet is trained for 60 epochs on WOMD.

---

> ### Comment · Reviewer_Ccz6 · 2023-08-14
> **Response to rebuttal**
>
> I appreciate the additional comparisons and clarifications provided by the authors. I understand that GoRela is not open-source and may not contain sufficient details to faithfully replicate the results, so I won't consider GoRela in comparisons.
>
> The central claim of the paper is efficiency. To achieve this, 2 components are proposed - KNARPE (K-nearest neighbor attention with relative pairwise encoding) and HPTR (hierarchical polyline transformer with asynchronous token updates). To verify this claim, I'd expect to see comparisons with the most relevant baseline, which is HiVT [70] (from my understanding). This is due to several reasons:
> - HiVT also uses pairwise relative pose encoding (to account for relative translation and rotation) in agent-agent, agent-lane, and global interaction modules (Sec. 3.3.1 & Sec. 3.3.2 in [70]).
> - HiVT consists of a transformer-based architecture that operates on vectorized inputs (polylines are also a form of vectorized inputs) and encodes both global and local pairwise information (Fig. 1 in [70]).
> - HiVT also considers motion prediction task and the results on inference speed in [70] seem to indicate that it can also run real-time with similar latency as HPTR (Sec. 4.3 and Table 5 in [70], although the numbers may not be directly comparable due to different settings).
> - The main difference between HPTR and HiVT is asynchronous token update.
>
> Due to these similarities, it is important to compare with HiVT (HiVT is published in CVPR 2022 and the code is publicly available). However, the experiments in the paper do not contain HiVT as a baseline. It'd be helpful to get some more insights into the differences between HPTR and HiVT and why is HiVT not considered in the experiments.

---

> > ### Author Response · Authors · 2023-08-16
> > **Answers to the questions raised by Reviewer Ccz6 regarding HiVT**
> >
> > Dear Reviewer Ccz6,
> >
> > Thank you very much for your comments.
> > We are glad that we have addressed the concerns you raised in the initial rebuttal phase.
> > Since here is the global response, we want to keep this thread short and concise.
> > We have also prepared a thorough answer in the thread specific to your review in order to address your new concerns regarding HiVT.
> >
> > Besides the asynchronous token update which you have pointed out, the additional fundamental differences between our method and HiVT have been briefly discussed in L93 of our paper.
> > In your individual thread, we will elaborate on this in more detail.
> > A short summary of that thread is as follows.
> >
> > 1. HiVT uses the agent-centric representation and the standard attention mechanism.
> > It does not share information among agents or across time steps.
> > By contrast, we utilize the pairwise-relative input representation and introduce a new attention mechanism.
> > Therefore, HiVT is not the most relevant baseline for our method.
> >
> > 2. HiVT reports performance only on AV1, which is outdated and has been replaced by AV2.
> > HiVT has been outperformed by many published methods, e.g. Wayformer and MultiPath++, by a large margin on AV1.
> > Since we have compared with Wayformer and MultiPath++ on the more up-to-date datasets (WOMD and AV2) and our performance is on par with MultiPath++, it makes little sense to consider the older and lower-performing HiVT as an additional baseline in our paper.
> >
> > 3. HiVT does not focus on efficiency and scalability.
> > As an agent-centric method, HiVT fundamentally suffers from poor scalability.
> > The latency of HiVT is close to that of our method, but HiVT considers a simpler dataset (AV1) and it uses fewer parameters (2.5M) and a smaller perceptive field (50 meters) in its implementation, all of which reduce latency but not the algorithmic complexity.
> > Therefore, the conclusion that HiVT is on par with our HPTR in terms of efficiency is not correct.
> >
> > ---
> >
> > To conclude, the three major contributions we have claimed in our paper are still valid and novel, as they are not presented in the HiVT paper.
> > For complete details, please refer to the discussion thread with Reviewer Ccz6.

---

> > > ### Comment · Reviewer_Ccz6 · 2023-08-20
> > > **Discussion about efficiency claims and comparison with baselines**
> > >
> > > The key ideas in HPTR architecture are related to pairwise relative representation, restricting the attention to K-nearest neighbors, using vectorized inputs (in the form of polylines), and asynchronous token updates in the transformer. Some of these ideas have been explored previously through different design choices:
> > > - pairwise relative representation (GoRela, HDGT, HiVT)
> > > - restricting the attention to local neighborhood (HiVT)
> > > - using vectorized inputs (HiVT)
> > > - efficient attention architectures (factorized attention, latent query attention in Wayformer)
> > >
> > > I agree that there are differences in architectural design choices between HPTR and baselines like HiVT & Wayformer. That is not an issue.
> > >
> > > The concern is related to verifying the central claim of the paper, which is efficiency. For this, I expect to see clear efficiency gains over baselines that use some combination of vectorized inputs, pairwise relative information, and/or efficient attention/transformer architectures. HiVT and Wayformer are quite relevant in these aspects and they also consider agent-agent, agent-lane, and temporal interactions in their architecture. This is why I think comparison to HiVT, Wayformer (factorized attention, latent query variants) are important.
> > >
> > > Looking at the results in Wayformer & HiVT papers, they report latency in the range of 30-60ms for different variants which is in the similar range as HPTR. I agree that these numbers may not be directly comparable and that is why a fair comparison is required to verify this. I think the results should be shown in the form of performance vs latency vs capacity plots (similar to Fig. 4,5,6 in Wayformer paper) while comparing to different baselines to show the benefits of HPTR.

---

> > > > ### Author Response · Authors · 2023-08-21
> > > > **Answers to the questions raised by Reviewer Ccz6 regarding efficiency**
> > > >
> > > > Dear Reviewer Ccz6,
> > > >
> > > > Thank you very much for your comments. We are glad that we have addressed your concerns regarding HiVT.
> > > > In the following we will answer your new questions regarding our efficiency claims and comparison with baselines.
> > > >
> > > > ---
> > > >
> > > > >The concern is related to verifying the central claim of the paper, which is efficiency.
> > > >
> > > > The new questions are based on this claim, but this claim is not precise.
> > > > As presented in our abstract (L4-5), the central claim of our paper is scalability in real-world, i.e. we investigate how to realize real-time motion prediction with streaming inputs on a real car surrounded by a large number of ($N=64$) agents.
> > > > This is different from the efficiency comparison in prior works such as HiVT and Wayformer, where just the offline inference latency is reported for a small number ($N\leq8$) of agents.
> > > >
> > > > ---
> > > >
> > > > >For this, I expect to see clear efficiency gains over baselines that use some combination of vectorized inputs, pairwise relative information, and/or efficient attention/transformer architectures. HiVT and Wayformer are quite relevant...
> > > >
> > > > We have reimplemented Wayformer and compared to it as discussed in Sec. 4.4 and Fig. 4 of our paper.
> > > > Our HPTR clearly outperforms Wayformer during the offline inference with $N\geq 16$ agents, as well as during the online inference with $N\geq 1$ agents.
> > > > The efficiency gain of our method increases significantly when the number of agent increases; this shows the good scalability of our method compared to the agent-centric methods such as Wayformer and HiVT.
> > > >
> > > > ---
> > > >
> > > > >Looking at the results in Wayformer and HiVT papers, they report latency in the range of 30-60ms for different variants which is in the similar range as HPTR.
> > > >
> > > > As mentioned before, there is a difference between efficiency and scalability.
> > > > Reviewer Ccz6 focuses on the efficiency comparison considering only 8 agents, while our paper focuses on the scalability, i.e. efficiency comparison considering a lot of agents.
> > > > As shown in Fig. 4 of our paper, the efficiency gain of our method is enormous as the number of agents increases.
> > > > Considering only 8 agents, our method does not outperform agent-centric methods during the offline inference (c.f. Fig.4 of our paper).
> > > > However, since we are solving a real-world problem, motion prediction on a real car has to run online and deal with a lot of agents, rather than doing offline inference with 8 agents.
> > > >
> > > > ---
> > > >
> > > > >I think the results should be shown in the form of performance vs latency vs capacity plots (similar to Fig. 4,5,6 in Wayformer paper) while comparing to different baselines to show the benefits of HPTR.
> > > >
> > > > The performance vs. latency and the performance vs. capacity figures (Fig. 4,5,6 in Wayformer paper) are used to investigate the possibility of reducing the model size (hence also the latency) while keeping the performance unaffected.
> > > > This trade-off is heuristically determined in our paper based on the analysis of Wayformer as well as considerations on the latency and computation.
> > > > In practice we set the total number of learnable parameters to roughly 15M for all ablation models and we make sure all ablation models have achieved reasonable performance.
> > > > Since our paper focuses on the scalability to large number of agents and the online inference, we believe Fig. 4 of our paper can support our arguments better compared to the performance vs. latency/capacity figures.
> > > > Using a different model size will not change the results presented in Fig. 4 of our paper.
> > > > As shown in our paper (Fig. 4) and prior works (e.g. Wayformer), agent-centric methods can achieve real-time motion prediction, but only when the number of considered agents is small.
> > > > The poor scalability of agent-centric methods cannot be solved by reducing the model size.
> > > >
> > > > ---
> > > >
> > > > We hope we have addressed all your concerns this time.
> > > > Since the discussion phase will end in a few hours, we might not have enough time to answer further questions.

---

> > > > > ### Comment · Reviewer_Ccz6 · 2023-08-21
> > > > > **Thank you for the clarification**
> > > > >
> > > > > - $N \leq 8$ setting is not entirely clear to me since the experiment settings seem to be different in baseline papers: [Wayformer](https://arxiv.org/pdf/2207.05844v1.pdf) mentions that max num of context agents = 64 (Table 3 in Appendix B) whereas [HiVT](https://openaccess.thecvf.com/content/CVPR2022/papers/Zhou_HiVT_Hierarchical_Vector_Transformer_for_Multi-Agent_Motion_Prediction_CVPR_2022_paper.pdf) mentions the average number of agents per scene close to 32 (Sec 4.3 Inference Speed). Maybe there are several differences in the protocol considered in HPTR and the baseline papers, which is causing all the confusion.
> > > > >
> > > > > - L288-289 mentions that the WF baseline is a reimplementation of the Wayformer with multi-axis attention and early fusion. Wayformer has different variants out of which factorized & latent query attention variants are more efficient than multi-axis attention. Comparison to these efficient attention variants is more desirable. Again, it is possible that important details might be missing in the text, which is causing all the confusion.
> > > > >
> > > > > Overall, I appreciate all the clarifications provided by the authors, which are quite useful in getting a better understanding of the paper.
> > > > >
> > > > > I have 2 suggestions regarding the text in the paper:
> > > > > - Clearly specify how the evaluation protocol differs from those used in baseline papers and why are those protocols not considered. This is important so that if the readers refer to the baseline papers, they should not have confusion regarding these important details.
> > > > > - Include a separate baseline subsection in Sec 4 to specify details about which baselines are considered for the efficiency analysis. The details should include similarities and differences between the baseline used in the evaluation and those present in the original papers (and also, the reason for differences should be included). This is important so that if the readers refer to the baseline papers, it should be clear to them which variant of the baseline is considered and if the results in the baseline papers could be compared to the current paper.
> > > > >
> > > > > In the absence of these details, it could raise concerns about the comparisons and cause confusion.

---

> > > > > > ### Author Response · Authors · 2023-08-21
> > > > > >
> > > > > > Dear Reviewer Ccz6,
> > > > > >
> > > > > > Thank you very much for your comments.
> > > > > >
> > > > > > Sorry for the confusion. The $N\leq 8$ we mentioned is the number of agents to be predicted.
> > > > > > In contrast to agent-centric methods, pairwise-relative methods such as GoRela and our HPTR do not differentiate between the context agents and the agents of interest.
> > > > > > The number of context agents in our experiments is always the same as the number of agents to be predicted.
> > > > > >
> > > > > > We choose the multi-axis Wayformer because it is the preferred solution in the Wayformer paper.
> > > > > > It is also the variant that performs the best on the WOMD leaderboard.
> > > > > > As discussed in the Wayformer paper, other variants may be faster but also perform worse.
> > > > > >
> > > > > > Following your suggestions we will make the experiment setup clearer and highlight the differences better.
> > > > > > If there’s not enough space in the main paper we will add the details to the appendix.

---

### Decision · Program_Chairs · 2023-09-21

**Decision:**

Accept (poster)

**Comment:**

The submission received three weak accept (WA-6), one borderline reject (BR-4) and one reject (R-3) recommendations from the reviewers. The authors’ responses seems to be convincing for the most reviewers engaged in discussion, three 6 ratings favour acceptance. After considering the paper, rebuttal and the reviews, the AC concurred with the acceptance recommendation. Please consider to incorporate all the additional responses & experiments in the final version. The AC also recommend the authors to consider acknowledging many missing seminal, but relevant, literature in the final version.